# Forge: Foundational Optimization Representations from Graph Embeddings

**Zohair Shafi**  *shafi.z@northeastern.edu*
*Khoury College of Computer Science*
*Northeastern University*

**Serdar Kadıoğlu**  *serdark@cs.brown.edu*
*AI Center of Excellence, Fidelity Investments*
*Department of Computer Science, Brown University*

**Reviewed on OpenReview:** *https://openreview.net/forum?id=7Uo1yRWUpo*

## Abstract

Combinatorial optimization problems are ubiquitous in science and engineering. Still, learning-based approaches to accelerate combinatorial optimization often require solving a large number of difficult instances to collect training data, incurring significant computational cost. Existing learning-based methods require training dedicated models for each problem distribution, for each downstream task, severely limiting their scalability and generalization. We introduce FORGE: **F**oundational **O**ptimization **R**epresentations from **G**raph **E**mbeddings, a framework that pre-trains a vector-quantized graph autoencoder on a large, diverse collection of mixed-integer programming (MIP) instances in an unsupervised manner, without relying on optimization solvers or optimal solutions. Vector quantization produces discrete code assignments that serve as a vocabulary for representing optimization instances. We evaluate FORGE in both unsupervised and supervised settings. In the unsupervised setting, FORGE embeddings effectively cluster unseen instances across problem domains and sizes. In the supervised setting, we fine-tune FORGE embeddings and show that *a single pre-trained model* helps predicting both the integrality gap for cut-generation and variable hints for search guidance across multiple problem and size distributions. In both tasks, we improve the performance of a commercial optimization solver and outperform state-of-the-art learning-based methods. Finally, we open-source our training code, pre-trained FORGE weights, and embeddings for multiple MIP distributions to foster further research in representation learning for optimization problems at `https://skadio.github.io/forge/`.

## 1 Introduction

Combinatorial Optimization (CO) problems are fundamental in science and engineering with applications spanning diverse domains, including logistics, energy systems, network design, and recommendations (Gabriel Crainic et al., 2021; Miller et al., 2023; Kadıoğlu et al., 2024). Traditionally, CO problems have been solved using carefully designed meta-heuristics and sophisticated optimization solvers. While effective, such classical methods demand significant domain expertise and computational resources, especially as problem size and complexity grow.

Recent advances in Machine Learning (ML) have introduced promising alternatives for solving CO problems. The approaches for ML-guided optimization fall mainly under two categories: 1) end-to-end models that predict solutions or objective without relying on solvers or meta-heuristics, and 2) hybrid methods which either 2a) replace computationally intensive solver components with learned models, e.g., learning to predict strong branching heuristics (Gasse et al., 2019), or 2b) guide meta-heuristics with learning from feedback (Cai et al., 2025b). For a comprehensive overview of ML-guided CO, we refer readers to the survey by Bengio et al. (2021b).

Despite their potential, learning-based methods face practical limitations. A significant drawback of learning approaches is their heavy dependency on offline training. Training is computationally costly, depends on carefully curating training datasets with desired properties and distributions of the underlying CO instances, and has limited generalization. Ironically, training depends on optimization solvers, that they try to accelerate, to create labeled datasets. This defeats the purpose of improving solving for challenge instances that optimization solvers cannot deal with today. Adapting learning-based methods to new distributions and domains remains a challenge, and hence, *designing a foundational optimization representation in an unsupervised fashion applicable to multiple optimization scenarios* is a much-needed alternative. This is exactly what we study in this paper.

Our main motivation is grounded in the exceedingly successful foundational methods in other modalities, such as text and image embeddings. This raises a natural question for optimization: can we leverage the abundance of publicly available mixed-integer programming (MIP) instances to develop a pre-trained, general-purpose foundational model for MIP representations that serves multiple optimization tasks, across varying problem domains and sizes?

The growing success of ML-based approaches for optimization problems makes this direction imminent. For example, Zhou et al. (2023) propose a meta-learning framework that generalizes across variants of vehicle routing problems of different sizes, but remains limited to routing. Similarly, Cai et al. (2025a) introduce a multi-task framework for backdoor prediction and solver configuration, however, the shared model is trained separately for each problem. Likewise, Li et al. (2025) propose an LLM-based evolutionary framework to generate diverse MIP problems, but is supervised, and requires a large number of pre-solved instances. While promising, the existing work either generalizes across multiple tasks but remains problem-specific, or scales across different sizes and variants but is task-specific, or remains dependent on solvers. We envision a foundational model that produces MIP embeddings generalizable across tasks, problems, and sizes, trained in an *unsupervised fashion*, without relying on solving hard optimization problems.

When we look at Natural Language Processing (NLP) and Computer Vision (CV), foundational models have emerged through unsupervised or self-supervised training, enabled by the abundance of data. CO problems, while also benefiting from publicly available datasets, span highly heterogeneous problem types (e.g., Set Covering vs. Combinatorial Auction) and exhibit significant variability within each problem. Most ML-based approaches to CO rely on Graph Neural Networks (GNNs), as proposed in (Gasse et al., 2019), which effectively capture local variable and constraint-level information but struggle to encode meaningful global structure. This limitation, rooted in the inherent locality bias of GNNs, is analyzed in detail by (Feng et al., 2025). As a result, while embeddings for other modalities such as text, image, and audio are now widespread, *no general-purpose instance-level embeddings exist for MIPs to date.*

With this vision in mind, we propose FORGE: **F**oundational **O**ptimization **R**epresentations from **G**raph **E**mbeddings. FORGE[1] is a step toward a foundational model designed to generate MIP embeddings through a pre-training framework that learns *structural representations at the instance level in an unsupervised manner*, using a broad distribution of MIP instances without requiring access to their solutions.

To achieve this, we incorporate two key ideas; one inspired by NLP and the other by CV. From NLP, we adopt the concept of a *vocabulary* to represent the latent space of optimization problems, enabling instance-level representations. From CV, we leverage *vector quantization* to preserve global information, addressing the limitations of GNN-based approaches in prior CO work. By extending these two crucial insights into the CO context, we propose a framework for CO where a single model can be pre-trained to operate across different problem types, and the *same pre-trained model* can further be fine-tuned for different optimization tasks. Concretely, we make the following contributions:

➪ **A Step Toward a Foundational Model for Optimization:** We propose FORGE[1], as step toward general-purpose foundational model for generating MIP embeddings (§3). FORGE captures both local and global structures critical to optimization. Unlike prior work, a single pre-trained FORGE model provides embeddings at multiple levels: instance-level representations (one vector per MIP instance), and fine-grained variable and constraint embeddings.

---

[1] https://skadio.github.io/forge/

➭ **Unsupervised Generalization:** In the unsupervised setting, FORGE embeddings cluster previously unseen instances across diverse problem types with high accuracy (§4).

➭ **Supervised Adaptability:** In the supervised setting, pre-trained FORGE embeddings can be fine-tuned on diverse downstream tasks using minimal additional data and a low-cost labeling strategy that avoids solving to optimality. We evaluate FORGE on two distinct tasks: estimating integrality gap for cut generation (§5.1) and predicting variables for search guidance (§5.2). Notably, *a single pre-trained FORGE model* is fine-tuned and applied across varied domains and problem sizes.

➭ **Solver Integration:** To enhance traditional optimization solvers, we integrate FORGE predictions into GUROBI (Gurobi Optimization, LLC, 2024), a state-of-the-art commercial solver, and demonstrate consistently lower primal gap across tasks and a wide range of problem domains and sizes.

➭ **ML Augmentation:** To enhance ML-guided optimization, we evaluate FORGE against (Li et al., 2025) for integrality gap prediction, and PS-Gurobi (Han et al., 2023) for search guidance, improving their performance on large sets of instances they were trained on, yet unseen by FORGE.

## 2 Background: Mixed-Integer Programming

Let us start with a brief background on Mixed-Integer Programming (MIP) that formulates combinatorial optimization problems of the form:

$$f(x) = \min\{c^T x \mid Ax \leq b, x \in \mathbb{R}^n, x_j \in \mathbb{Z} \ \forall j \in I\}$$

where $f(x)$ is the objective function, and $A \in \mathbb{R}^{m \times n}, b \in \mathbb{R}^m, c \in \mathbb{R}^n$, and the non-empty set $I \subseteq 1, ..., n$ indexes the integer variables. The Linear Programming (LP) relaxation of a MIP $x_{lp}$ is obtained by relaxing integer variables to continuous variables, i.e., by replacing the integer constraint $x_j \in \mathbb{Z} \ \forall j \in I$ to $x_j \in \mathbb{R} \ \forall j \in I$. The LP relaxation is an essential part of the branch-and-bound algorithm for providing bounds. The integrality gap measures how much worse the optimal solution of the LP relaxation when compared to the optimal solution of the original MIP.

## 3 Forge: Unsupervised Representation Learning for MIPs

MIP instances are typically represented as a bipartite graph between variables and constraints augmented with node features (Gasse et al., 2019; Ferber et al., 2022; Yau et al., 2024; Chen et al., 2023b). This is then followed by training a GNN in a supervised fashion for a specific downstream task on a certain problem class. For example, in Han et al. (2023), the GNN is used for predicting variables for warm-starts trained on Set Cover (SC) and Independent Set (IS) problems. Similarly, in Li et al. (2025), a GNN is used for predicting the integrality gap. Numerous variants follow this template (see related works in §6 and Appendix A.9). Notice that, *all of these methods require supervision* and do not yield a general-purpose MIP embeddings at the instance level. Taking this a step further; *our goal is to learn the structure of MIP instances in an unsupervised manner.*

Figure 1 presents our overall architecture, which is composed of these main building blocks:

**A) MIP-to-BP:** Given a MIP instance, we start with its bipartite (BP) representation and node features. Each node in this bipartite graph represents a constraint or a variable, with edges indicating which variables are part of which constraints. Each node is associated with node features and each edge is weighted by the coefficient of the variable in the constraint. Node features are typically extracted from the internal branch-and-bound search tree when solving an instance. For example, there are 18 input node features used in (Gasse et al., 2019). We do not attempt to solve or depend on the solution of the instance. Instead, FORGE only uses the basic properties of the input instance. For each constraint node, we introduce 4 features composed of its sense (i.e., $>$, $<$ or $=$) and the RHS value. For each variable node, we introduce 6 features composed of its type (integer, binary, continuous), upper/lower bound, and the coefficient in the objective function. In total, for each node, we obtain a vector of size 10, padded with zeros accordingly based on node type (Figure 1-A).

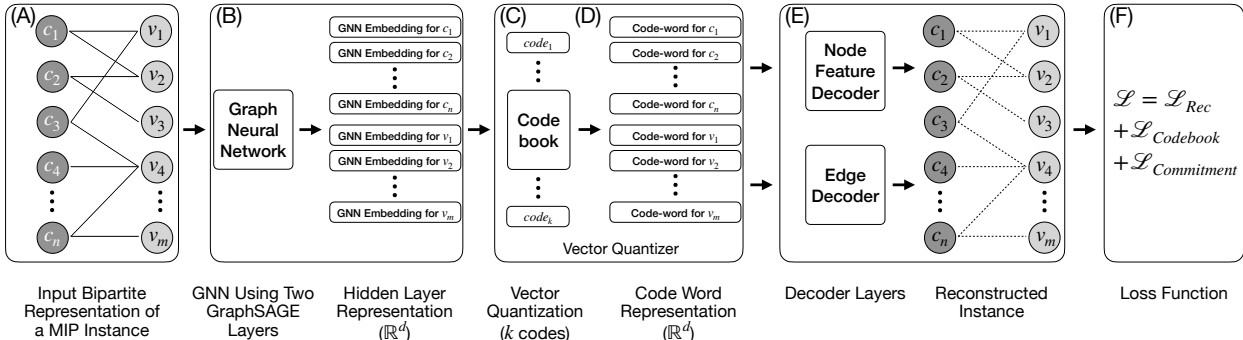

Figure 1: FORGE: Our approach for learning MIP embeddings without supervision. Starting with the bipartite representation and its GNN embedding (A-B), FORGE uses a vector quantized graph autoencoder (C-D) to reconstruct node features and edges (E). It is pretrained across a diverse set of problems and sizes to learn generic MIP representations without dependency on optimal solutions.

**B) BP-to-GNN:** This bipartite graph with 10-dimensional input node features is passed into a GNN, akin to previous works, to generate embeddings for each constraint and variable node. More specifically, FORGE uses two GRAPHSAGE (Hamilton et al., 2017) layers that project each input node into a $d$ dimensional embedding space. As discussed earlier, while GNNs are good at capturing local variable- and constraint-level information, they struggle to capture meaningful global information at the instance level due to their inherent locality bias (Feng et al., 2025). Preserving global structure is important in CO problems, especially to generalize across problem types (Figure 1-B).

**C) Vector Quantized Codebook:** To preserve global structure, we introduce a vector quantized codebook with $k$ discrete codes. These codes act as a 'vocabulary', akin to language models, across MIP instances of various domains and difficulties, thereby preserving global structure. The design follows the approaches developed in computer vision (Van Den Oord et al., 2017; Yu et al., 2022; Lee et al., 2022) and the structure-aware graph tokenizer extension in Yang et al. (2024) (Figure 1-C).

**D) GNN-to-CW:** GNN embeddings are passed into a vector quantizer which consists of a codebook with $k$ codes. The codebook maps each variable and constraint node to a discrete code. Each code is then mapped into a $d$ dimensional codeword (CW), producing CW representations for constraints and variables, aligned with the dimensionality of the hidden GNN layers (Figure 1-D).

**E) CW-to-BP:** We use the CW corresponding to each constraint and variable node to reconstruct the original bipartite representation of the MIP instance. These codewords are passed into a linear node feature decoder and a linear edge decoder to reconstruct the input bipartite graph. By doing so, we obtain an *unsupervised method* that learns from the structure of MIP instances (Figure 1-E).

**F) Loss Function:** Our loss function minimizes edge reconstruction loss, node feature reconstruction loss, and losses related to the vector quantization (Van Den Oord et al., 2017). Concretely, the loss function is:

$$\mathcal{L} = \mathcal{L}_{Rec} + \mathcal{L}_{Codebook} + \mathcal{L}_{Commitment} \tag{1}$$

where given $N$ nodes, input node feature $v_i \ \forall i \in N$, the adjacency matrix $A$ and a matrix $\hat{X}$ composed of reconstructed input features $v_i$, the reconstruction loss, $\mathcal{L}_{Rec}$, the codebook loss, $\mathcal{L}_{Codebook}$, and commitment loss, $\mathcal{L}_{Commitment}$ are given by:

$$\mathcal{L}_{Rec} = (A - \hat{X}\hat{X}^T)^2 + \frac{1}{N}\sum_{i=1}^{N}(\hat{v}_i - v_i)^2 \tag{2}$$

$$\mathcal{L}_{Codebook} = \frac{1}{N}\sum_{i=1}^{N}\|sg[h_i] - cw_i\|_2^2 \tag{3}$$

$$\mathcal{L}_{Commitment} = \frac{\alpha}{N}\sum_{i=1}^{N}\|sg[cw_i] - h_i\|_2^2 \tag{4}$$

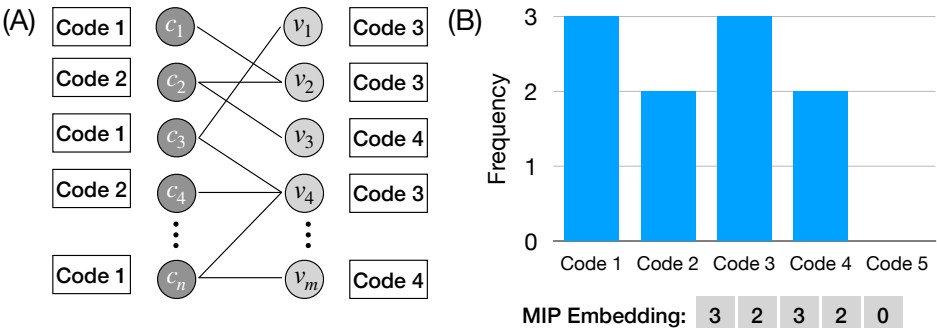

Figure 2: (A) Each node in the bipartite graph representation of the MIP instance is assigned a discrete code. (B) The distribution of these assigned codes yields the embedding of the MIP instance. The embedding dimension is the size of the codebook, i.e., $vocabulary = |codebook|$. This example uses 5 codes leading to the MIP embedding $\vec{emb} = [3, 2, 3, 2, 0]$.

In the loss function, $sg[.]$ is the stop-gradient operator, $h_i$ is the hidden layer representation of node $i$ after the GNN forward pass and $cw_i$ is the codeword corresponding to the code that node $i$ has been assigned. Intuitively, the codebook loss in Eq. 3 can be interpreted as k-means clustering, where the codewords $cw_i$ (akin to cluster centroids) are moved closer to the node embeddings $h_i$ and the node embeddings $h_i$ are fixed in place due to the stop-gradient operator. Conversely, the commitment loss in Eq. 4 fixes the codewords $cw_i$ using the stop-gradient operator, and instead, moves the embeddings $h_i$ towards the codewords. The hyperparameter $\alpha$ weighs the importance of the commitment loss.

Once FORGE is trained in this unsupervised manner across a corpus of MIP instances, we obtain:

**Local Constraint & Variable Representations**: Each node in the bipartite graph is assigned a discrete code which is mapped to a codeword. These become the constraint and variable embeddings.

**Global MIP Instance Representation:** We leverage the distribution of codes at the instance level. Each instance is represented with an embedding of vocabulary size, $|codebook|$, where each value indicates the frequency of the corresponding code within the instance. Figure 2 shows an illustrative example with a codebook of size 5 (for brevity) and the resulting MIP embedding $\vec{emb} = [3, 2, 3, 2, 0]$

Beyond unsupervised pre-training, for MIP-specific downstream tasks such as integrality gap prediction, guiding the search, and solver configuration, we can fine-tune *the same pre-trained embedding* using a small number of cheaply labeled data for radically *different tasks across different problems.*

In the following, we start with an initial investigation of the effectiveness of FORGE embeddings when clustering unseen instances across various problem domains (§4), and then explore how they can enhance solving MIP instances via predicting the integrality gap (§5.1) and guiding the search (§5.2).

## 4    Initial Analysis of Forge Embeddings

We start our initial analysis with a comparison of FORGE embeddings in clustering unseen instances against two (ablation) baselines. In Appendix A.1, we also consider hand-crafted MIP features for clustering Hutter et al. (2014); Kadıoğlu et al. (2010). Our main finding is that these static features, that have been engineered carefully over decades and are solver-dependent, perform well in cluster while our unsupervied, pretrained FORGE model recovers their effectiveness as detailed next.

We investigate both the accuracy (quantitative) as well as visual inspection (qualitative) of the clustering. To do so, we train FORGE on a set of MIP instances from MIPLIB (Gleixner et al., 2021), and test it on unseen instances from the validation and test category of Distributional MIPLIB (D-MIPLIB) (Huang et al., 2024). We repeat this experiment with another unseen benchmark, strIPlib (Bastubbe et al., 2025) in Appendix A.2 with similar findings.

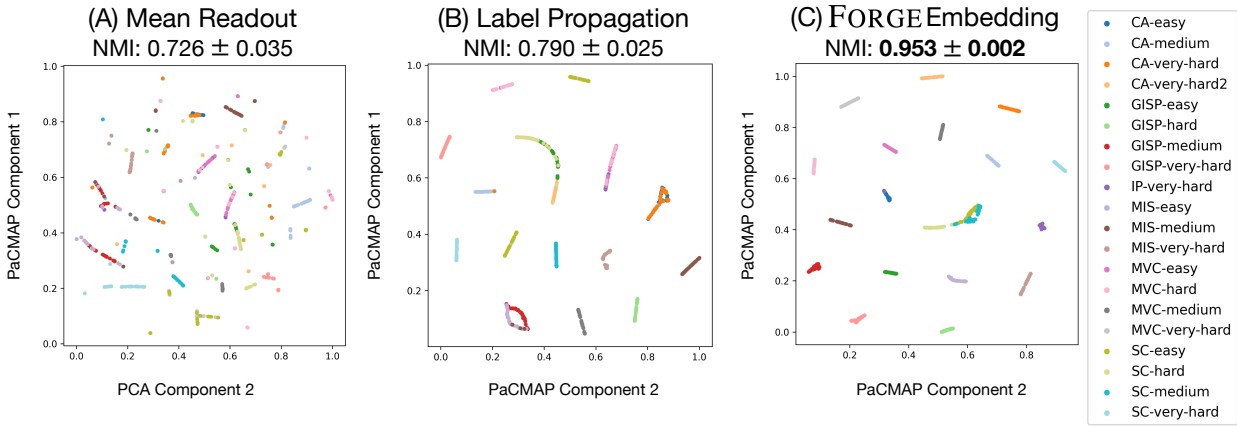

Figure 3: Visualization of MIP embeddings and NMIs from (a) the mean readout in FORGE, (b) two-hop label propagation of input node features, and (c) the distribution of discrete codes of FORGE.

While MIPLIB is a mixed dataset, D-MIPLIB and strIPlib are categorized into different problem classes and difficulties. This serves as a ground truth label when we treat each problem class as a cluster to evaluate our embeddings.

**Training:** We use 600 instances from MIPLIB, sorted by size to ensure the resulting bipartite graphs fit on GPU memory. For additional training data, we generate two instances from each MIPLIB instance by randomly dropping 5% and 10% of constraints (note that dropping constraints only relaxes the problem). In total, we obtain 1,800 MIP instances to train FORGE. Due to GPU memory, we had to skip ∼20 of these instances from MIPLIB with total number of nodes greater than 150K during training. We use two GraphSage layers with $d = 1024$ dimensions and a *codebook* with $k = 5000$ codes, the size of the vocabulary. We conduct an ablation study on the codebook size in Appendix A.3.

**Testing:** We evaluate FORGE on clustering 1,000 instances from D-MIPLIB categorized into 21 domain-difficulty pairs. These include Set Cover (SC easy, medium, hard), Max. Independent Set (MIS easy, medium), Min. Vertex Cover (MVC easy, medium, hard), Generalized Independent Set (GIS easy, medium, hard, very-hard, very-hard2, ext-hard), Combinatorial Auction (CA very-easy, easy, medium, very-hard, very-hard2), Item Placement (IP very-hard) and Maritime Inventory Routing (MIRP medium), covering a broad spectrum of problem domain, sizes and difficulty levels.

**MIP Embeddings:** These 1,000 unseen instances are passed into the pre-trained FORGE model to generate one embedding per MIP instance. For comparison, we consider two alternatives. As a baseline, we use the Mean Readout of the GNN embeddings within the trained FORGE model. This generates the embedding of an instance by averaging all node features from the GNN (Fig. 1-B). This behaves as an ablation for FORGE *without* vector quantization but with trained GNN embeddings. Given the weakness of GNN embeddings in capturing global structure, we expect this to perform poorly. Alternatively, starting from the 10-dimensional static node features of the bipartite graph (Fig. 1-A), we run two-hop label propagation and average the resulting node vectors (Zhu & Ghahramani, 2002). While static instance descriptors is uninformative, since within each category every instance has identical statistics (# of vars, constraints, etc.), a comparison with hand-crafted and solver-dependent MIP features from (Hutter et al., 2014) is in Appendix A.1.

**Clustering Visualization:** Figure 3 visualizes these embeddings vectors projected into 2D using PaCMAP (Wang et al., 2021). Each dot is an instance colored by its category. As anticipated, the Mean Readout method loses the global structure, resulting in mixed clusters (Fig. 3-A). The Label Propagation captures the structure better, although some mixing of problem types and difficulty levels remains (Fig. 3-B). **Forge embeddings performs the best, with distinct clusters for each problem** (Fig. 3-C). Furthermore, within each domain, a clear gradient from easy to hard problems is evident. Recall that, FORGE is *not trained* on these instances; but on MIPLIB only.

**Clustering Accuracy:** For quantitative evaluation, we run k-means with 21 clusters, expecting one cluster for each category. We calculate the normalized mutual information score (NMI) between the ground truth categories and clusters, averaged over 10 runs of k-means, for each method. If the predicted clusters are identical to the original categories, the NMI score is 1.0, and 0.0 otherwise. As qualitatively observed from the 2D visualization, the Mean Readout performs the poorest with an NMI score of 0.726. This is likely due to over-smoothing of averaging dense GNN embedding vectors across a large number of constraints and variables. In contrast, Label Propagation performs better with an NMI score of 0.790. This method operates directly on the sparse input features of the bipartite graph, hence avoiding the over-smoothing problem encountered in GNNs. **The best performance is achieved by Forge with an NMI score of 0.953.** Interestingly, the mean readout embeddings, which perform poorly, is based on the the same pre-trained GNN backbone in FORGE. By utilizing the distribution of the discrete codes of constraints and variables, FORGE circumvents the over-smoothing issue and captures the global structure of unseen MIP instances. As a proof of concept, we apply vector arithmetic on these MIP embeddings, drawing inspiration from language models and the well-known King - Man + Woman ≈ Queen example (Ethayarajh et al., 2018), to evaluate Vertex Cover - Cover + Packing ≈ Independent Set, and other combinations, in Appendix A.4.

## 5 Experiments

So far, we only demonstrated that FORGE embeddings reliably cluster unseen instances from diverse problem classes. Our ultimate goal is to improve MIP solving, hence we now shift to supervised evaluations. We design the next set of experiments on downstream tasks that (1) provide utility to enhance solving MIPs, (2) commonly applied in the literature to enable a fair comparison, (3) radically different from each other, whereby we use *the same FORGE model* to validate its general applicability across tasks, problems, and sizes, and (4) agnostic to the underlying MIP solver, i.e., we do not depend on internal access to specific solver procedures within the branch-and-bound tree.

We therefore study two fundamentally different downstream tasks: predicting the integrality gap of a given MIP instance, as also studied in (Li et al., 2025), and predicting variables for search guidance, as also studied in (Han et al., 2023). Despite being unrelated, both tasks serve as primal heuristics to speed-up MIP solving by obtaining better solutions faster. The integrality gap is used to generate a pseudo-cut added to the original problem formulation to tighten its bound at the beginning of the search, whereas variable guidance is used to provide hints to the solver during search. Preliminary results on predicting solver configuration (Cai et al., 2025a) and transfer learning from optimization to satisfaction representations (Pal & Kadıoğlu, 2026) are in Appendix A.5 and Appendix A.6.

The critical aspect of these experiments is that we use the **same pre-trained Forge model** to obtain general-purpose MIP embeddings that are then fine-tuned on a small number of cheaply labeled data to learn prediction heads for **completely different tasks**, as shown in Figure 4-A. An analogy from NLP is to generate pre-trained word embeddings using a foundational model, and then to fine-tune prediction heads for entity extraction in a specific domain, e.g., finance, using a small set of labels. Our goal is to revive the success of foundational models from NLP and CV in the context of CO.

**Training the Foundational Forge Model**: The training for foundational FORGE is identical to the previous configuration used in clustering, with the only difference being the expanded dataset: we now train on both 1,800 MIPLIB instances and 1,050 D-MIPLIB instances. In total, FORGE is trained on a corpus of 2,850 MIP instances using a model with 3.25 million parameters. We again use two GraphSage layers with $d = 1,024$, and $k = 5000$, with more details on our experimental setup in Appendix A.7.

### 5.1 Task - I: Predicting the Integrality Gap for Pseudo-Cut Generation

**Integrality Gap Prediction:** As briefly mentioned in (§2), the integrality gap measures the ratio between the optimal solution of the LP relaxation and the optimal solution of the original integer program. Intuitively, this gap quantifies the quality of the approximation offered by the LP relaxation. A smaller gap means the LP relaxation is a good approximation and is close to the value of the best integral solution. Here, we are interested in predicting the integrality gap of a given MIP instance, without solving for its optimal integral solution value, as also studied in (Li et al., 2025).

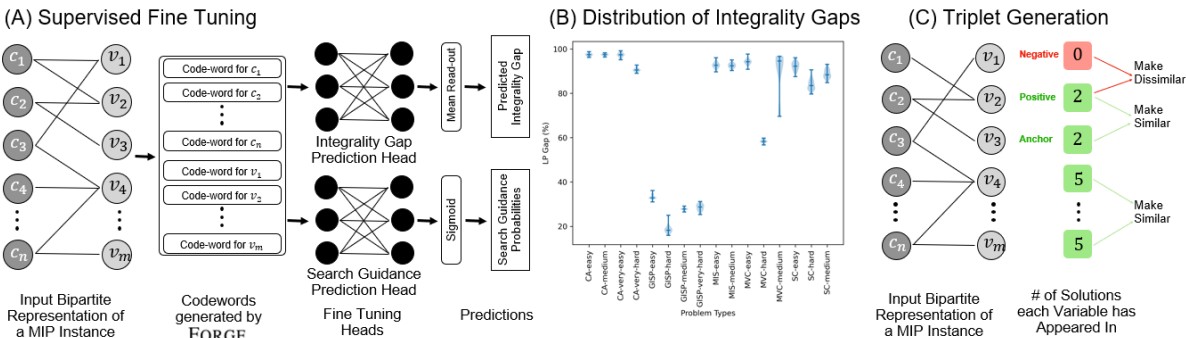

Figure 4: (A) Supervised fine-tuning flow for FORGE. (B) Distribution of integrality gap across problem types. (C) Triplet generation for search guidance by grouping positive/negative variables.

**Pseudo-Cut Generation:** If we can predict the integrality gap, then we can generate a *pseudo-cut* and add it as an additional constraint to the model to immediately bound the optimal objective value from the root node LP relaxation. Note that, this cut is not guaranteed to be a valid cut at all times, hence it is a *pseudo-cut*. If the integrality gap prediction is incorrect, it risks over (or under) estimating the best objective value. This makes integrality gap prediction a challenging problem.

**Training Instances:** We do not expect this task to generalize between problem classes and/or sets of varying complexities of a given problem class. For instance, there is no reason for an LP gap of 70% to be the same between easy vs. hard SC instances. Figure 4-B shows the distribution of integrality gap across different categories, and as expected, the integrality gap can be anywhere from 5% to 95% with wide distributions in between. As such, there is no magic constant that one could use heuristically at all times, which makes integrality gap prediction a deliberate learning task. For training, we only consider CA (very-easy, easy, medium), SC (easy, medium, hard), and GIS (easy, medium, hard) with 50 instances for each. In total, we obtain 450 training instances. This is a considerably smaller training set than initially used for pre-training FORGE embeddings.

**Label Collection:** To generate training labels, each training instance is solved using GUROBI with 120s time limit. Note that, this does not require solving instances to optimality, which can be costly. The numeric label is defined as the ratio between the integer solution at the time-out and the the LP relaxation. As previously noted, overestimating the integrality gap (in minimization problems) can lead to suboptimal outcomes as the solver may terminate prematurely. In contrast, underestimating the gap is generally acceptable, as it does not compromise solution quality. To account for this asymmetry, we adopt a conservative labeling strategy by enforcing a timeout during label collection. This approach might result in underestimating the true integrality gap, particularly for harder instances. By intentionally biasing toward underestimation, we reduce the risk of the model overestimating the gap, thereby enhancing the reliability of the predicted cut.

**Supervised Fine-Tuning:** Given this small and cheaply labeled data, a dense prediction head is added to the pre-trained FORGE model, as shown in Figure 4-A, that takes codewords assigned to each node as input and outputs a real number using mean readout across all nodes. As a regression task, this is trained with the mean absolute error loss in an end-to-end manner.

**Test Instances & Setup:** We use the fine-tuned FORGE to predict the integrality gap of 50 very-hard instances each of CA, SC, GIS, and MVC. Our fine-tuning does not include 'very hard' category, and MVC is entirely unseen in fine-tuning. Given the prediction, a pseudo-cut is generated by adjusting the initial LP relaxation objective and incorporating into the original formulation as an additional constraint. This enforces the integral objective to exceed (or fall below) the generated pseudo-cut.

**Prediction Accuracy:** We measure the deviation in mean absolute error between the known integrality gap and the gap predicted by FORGE. On these very-hard test instances, FORGE achieves a deviation of 15.42%, 13.55%, 12.03% and 19.077% for CA, SC, GIS, and MVC, respectively. As an ablation, training for this task from scratch, and **not using pre-trained Forge as starting weights worsens the error by ∼33%** on average across all categories. This highlights the importance of unsupervised pre-training to capture transferable structural patterns across diverse MIP instances.

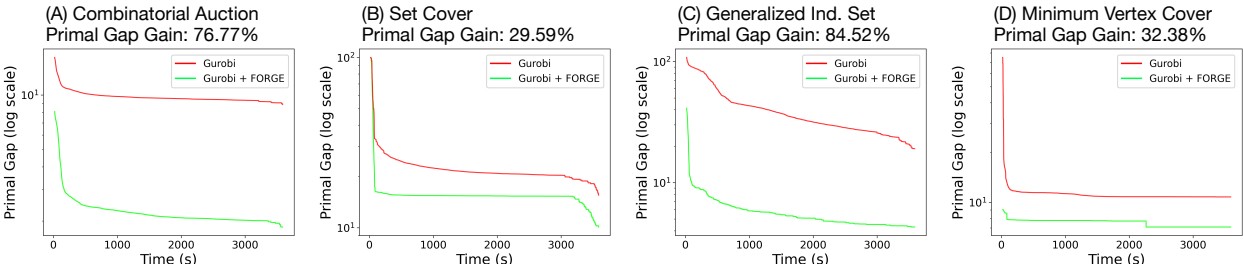

Figure 5: Gurobi vs. Gurobi + FORGE Pseudo-Cuts. Each subplot shows the primal gap (the lower, the better) averaged across 50 very-hard instances in each problem.

**Comparison with Commercial MIP Solver:** We compare the commercial GUROBI solver on these very-hard instances with and without our predicted pseudo-cut. Figure 5 shows the primal gap averaged over 50 instances of each problem with 3600s time limit. Without exception, across all problem types, the use of pseudo-cuts generated by FORGE consistently results in better primal gaps. The solver improves the gap early in the search, and our pseudo-cuts, make these gains immediately more pronounced. **The performance gains in the primal gap reach up to 85%.** Recall that fine-tuning for this task only needed 50 instances per problem type, not even including MVC in fine-tuning, and label collection had no dependency on optimal solutions.

**Comparison with SOTA ML:** To further evaluate generalization across problem types and sizes, we compare against the setup in Li et al. (2025), where a GNN is trained on 38,256 instances from 643 generated problem types and tested on 11,584 instances from 157 problem types. FORGE is used as is, *without any additional training*, and tested on 17,500 previously unseen instances from 400 generated problem types. **Forge achieves a mean deviation of 18.63% in integrality gap prediction**, improving over the 20.14% deviation reported in (Li et al., 2025).

## 5.2  Task - **II: Guiding the Search for Optimal Solutions**

Our next experiments extends the evaluation in two important dimensions. First, the previous task evaluates the *global* FORGE representations at the instance level while our next task evaluates the *local* FORGE representations, specifically the variable embeddings for search guidance. The idea is to fine-tune FORGE, on a smaller dataset with a labeling strategy that does not depend on solving to optimality, and provide variable hints to the GUROBI solver. Second, the previous task integrated with the solver in a single-shot manner, once at the root node as a single cut, whereas the next tasks helps guiding the solver throughout the branch-and-bound search. In principe, one can use FORGE for both at the same time, pseudo-cut at the root node and guide the search but here we control the study of effectiveness in isolation.

To provide search guidance, a naive method of predicting variables that are likely to be part of the solution is to treat it as a binary classification problem and use binary cross entropy (BCE) loss. However, this poses challenges due to the large class imbalance where most variables are not part of the solution. We address this issue by using a combination of BCE and triplet loss as follows.

**Training & Labeling:** We collect 100 instances from CA (easy, medium), SC (easy, medium, hard) and GIS (easy, medium) for a total of 700 training instances. Each instance is solved using Gurobi to find a pool of five feasible solutions within five minutes. As in the previous task, optimality is *not required* for labeling.

**Supervised Fine-Tuning:** Given five feasible solutions, variables that never appear in any solution is marked as 'negative' and variables that appear in a solution at least once is marked as 'positive'. FORGE is fine-tuned using a combination of binary cross-entropy (BCE) and triplet loss. For BCE, we add a dense prediction head to pre-trained FORGE as in integrality gap prediction (Figure 4-A). In parallel, we use the standard triplet loss (Schroff et al., 2015), where variables that appear in the *same number of solutions* are treated as 'positive' and 'anchor' pairs (Figure 4-C).

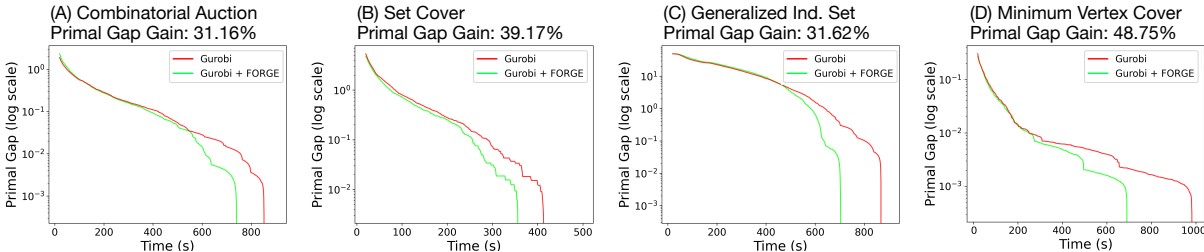

Figure 6: Gurobi vs. Gurobi + FORGE Search Guidance. Each subplot shows the primal gap (the lower, the better) averaged across 50 medium instances in each problem.

The triplet loss is given by:

$$L(a, p, n) = max\{d(a_i, p_i) - d(a_i, n_i) + margin, 0\} \tag{5}$$

In the triplet loss, $a$ is the 'anchor' node, $p$ is the 'positive' node, $n$ is the 'negative' node and $d$ is a distance function (euclidean distance in our case). This triplet loss aims to minimize the distance between the 'anchor' and 'positive' nodes while ensuring the 'negative' node is at least 'margin' distance away from the 'anchor' node (margin is set to two in our case).

A key challenge of triplet loss is finding good negative nodes, as picking trivially negative nodes does not aid learning. Our pre-trained FORGE helps circumvent this issue: for every positive/anchor pairs of variables, we select the negative variable that is *closest* to the anchor variable in the unsupervised FORGE embedding space. We fine-tune FORGE to minimize the sum of the BCE and triplet losses, weighted equally. One loss minimizes the binary labeling error and the other loss minimizes the distance between the embeddings of the 'anchor' and 'positive' variables while ensuring the 'negative' variable is at least 'margin' distance away from the 'anchor' variables

**Test Instances & Setup:** We test on 50 medium instances from each of CA, SC, GIS, and MVC. Again, MVC is unseen in fine-tuning. We use our fine-tuned FORGE to predict the likelihood of variables to appear in the solution. For search guidance, we begin with a feasible solution found by Gurobi within 1s. Variables appearing in this solution serve as *anchors*. The neighbors of the positive anchors within a fixed radius in the embedding space, that are also in the top-decile of the FORGE prediction head, are hinted to the solver for inclusion. Conversely, the neighbors of the negative anchors that are in the bottom-decile of the FORGE predictions are hinted to the solver for exclusion. This strategy exploits our training objective that optimizes for variable prediction using BCE loss and clustering of positive and negative variables using triplet loss.

**Comparison with the Commercial MIP Solver:** We compare the performance of GUROBI solver on these test instances with and without our search guidance introduced into the model as solver hints on variables. Figure 6 shows primal gaps averaged over 50 instances for each problem under a 3600s time limit. As in the previous experiments, the commercial solver, when powered by the search guidance from FORGE, achieves **consistently better primal gaps (up to 48% improvements)** and **converges to optimal solutions significantly faster (up to 35% speed-ups)**. In short, FORGE makes Gurobi faster and better on all tested problems demonstrating the practical utility of FORGE in accelerating MIP solving.

**Comparison with SOTA ML:** As a final experiment, we test the ability of FORGE to augment not only a MIP solver but also other ML-methods. For this, we augment the SOTA ML method, PS-Gurobi (Han et al., 2023), which also studies search guidance for GUROBI in the form of a predict and search framework. PS-Gurobi itself already demonstrates strong performance against previous learning-based approaches such as Neural Diving (Nair et al., 2020). We again use our pre-trained FORGE embeddings as-is and concatenate variable and node embeddings of PS-Gurobi with our *unsupervised embeddings* (after PCA to reduce to 64 dimensions to fit into to the architecture of PS-Gurobi). We then retrain PS-Gurobi from scratch with and without the concatenated embeddings. Since we retrain PS-Gurobi, we follow the common subset of problems, CA and GIS, that are used in PS-Gurobi experiments for which we are given best PS-Gurobi settings.

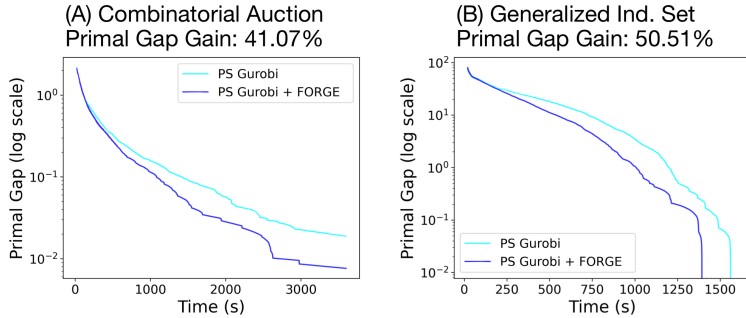

Figure 7: PS-Gurobi vs. PS-Gurobi + FORGE Search Guidance. Each subplot shows primal gap (the lower, the better) averaged across 50 medium instances in each problem.

As shown in Figure 7, augmenting PS-Gurobi with our FORGE embeddings yields significant improvements: **over 40% reduction in primal gap** on average for CA instances, and **over 50% reduction** on average for GIS instances. Additional randomized ablations to control for the increased model capacity from higher-dimensional inputs are provided in Appendix A.8.

# 6 Related Work

We cover immediately relevant work here and elaborate further in Appendix A.9. In terms of unsupervised learning, our model is closest to unsupervised approaches studied in (Sanokowski et al., 2024; Karalias & Loukas, 2020; Bu et al., 2024). However, these works aim to reformulate the discrete, combinatorial objective of *specific problems* into a differentiable one to *learn a solution* using gradient descent in an end-to-end manner. FORGE is different as it learns the *structure of the instance* in an unsupervised manner. This gives us the ability to represent *any MIP instance* off-the-shelf with a single pre-trained model. Other successful ML-based optimization methods show that generalization is possible, e.g., (Cai et al., 2025a) introduces multi-task learning using a shared model for predicting both backdoors and solver configuration, albeit trained per problem type. Problem-specific works such as (Zhou et al., 2023) shows generalization across variants of vehicle routing and (Shafi et al., 2025) for variants of set covering problems. The existing work either generalizes across multiple tasks but remains specific to one problem, or scales to different sizes and problem variants but is specific to one task, or remains supervised. FORGE is the first to generate MIP embeddings that generalize across multiple tasks, different problems and sizes, and trained in an *unsupervised fashion*, without dependency on solutions. Regarding downstream optimization tasks, (Li et al., 2025) addresses integrality gap prediction and (Han et al., 2023) focuses on search guidance, both of which we evaluate and compare against. In both cases, with minimal additional training and cheap labeling strategy, FORGE consistently matches and improves their performance. There exist other important downstream tasks, such as predicting solver parameters (Hosny & Reda, 2024), learning to branch (Khalil et al., 2016; Liberto et al., 2016), node selection (He et al., 2014a), and cut selection (Paulus et al., 2022). Beyond MIPs, meta-heuristics (Cai et al., 2025b), constraint satisfaction (Tönshoff et al., 2023), and SAT (Duan et al., 2022) show benefits from learning-based approaches. While most of these works, including ours, is based on GNN representations (Gasse et al., 2019), (Drakulic et al., 2024) avoids GNNs, and instead, uses mixed attention over edge and node matrices.

# 7 Limitations & Future Work

We present FORGE, a novel unsupervised framework for learning structural representations of optimization problems at the instance, variable, and constraint levels, without requiring access to solvers or ground-truth solutions. Inspired by NLP and CV, FORGE introduces a discrete vocabulary of optimization codes employing vector quantization to capture global structure.

A single pre-trained FORGE model clusters unseen instances from diverse benchmarks and generalizes across two distinct optimization tasks on several problem domains with varying difficulty levels. These embeddings integrate seamlessly into both a commercial solver and state-of-the-art ML pipelines, consistently yielding measurable performance improvements. Our study is subject to several limitations and opens the door for promising directions for future research:

- **Scale:** FORGE is compact in size (3.25M parameters trained on ∼2.8 instances), and is much smaller compared to large-scale models in other domains. In principle, it is feasible to train our framework on all publicly available and synthetically generated MIP instances.

- **Interpretability:** The semantics of the learned optimization vocabulary remain completely unexplored. Our preliminary evidence suggests that certain codes capture local structure, such as cliques of variables and constraints, enabling generalization to larger problems.

- **Solver Integration:** Current experiments are one-shot, using embeddings to generate a pseudo-cut or guide the solver once. Extending this to operate throughout the branch-and-bound tree could enable tighter integration with the solving process for further improvements.

- **Downstream Tasks:** Many other important optimization tasks remain unexplored. FORGE embeddings can be leveraged for warm-starts, variable selection, node selection, cut selection, solver configuration, and portfolio construction, among others.

- **Generalization:** The underlying principles of FORGE may extend beyond optimization to constraint satisfaction problems as well as from complete branch-and-bound search to incomplete search methods and meta-heuristics. Our initial work (Pal & Kadıoğlu, 2026) shows early promising results on transfer learning from pretrained FORGE optimization models to unseen Boolean Satisfiability (SAT) instances. This opens yet an exciting future direction: to train a heterogeneous FORGE model on a mixture of MIP *and* SAT instances, enabling hybrid, multi-domain pretraining for combinatorial problems analogous to multi-modal vision language (VLM) models.

To enable these future directions and support reproducible research, we open-source[2] our datasets, training pipelines, pre-trained and fine-tuned FORGE models, readily available MIP embeddings across problem distributions from MIPLIB, D-MIPLIB, and strIPlib for the community.

## 8 Acknowledgments

The authors would like to thank Modal for their generous support through the provision of academic credits and computational infrastructure, which were instrumental in training the FORGE model used in this research.

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

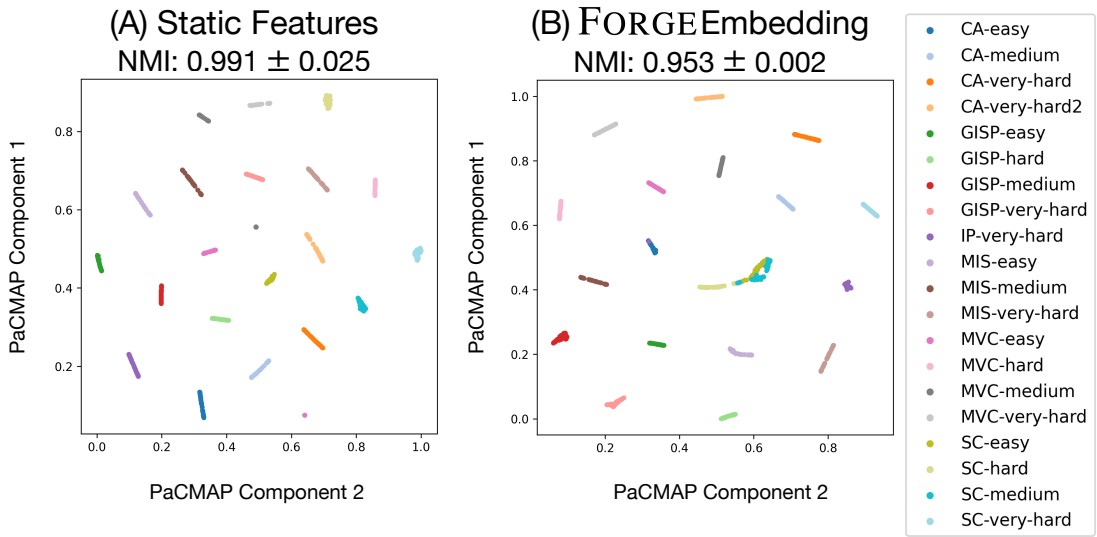

Figure 8: Clustering performance (Normalized Mutual Information) of FORGE using 10 input features compared to the 100 static features proposed by Hutter et al. (2014). Despite operating on an order of magnitude fewer input features, FORGE achieves comparable NMI, indicating that its learned representations capture sufficient structural information to match a carefully hand-engineered and solver dependent feature set.

# A   Appendix

## A.1   Comparison with Static MIP Features

A substantial body of prior work has explored hand-crafted feature sets for characterizing MIP instances Hutter et al. (2014); Kadıoğlu et al. (2010). Notably, Hutter et al. (2014) propose approximately 100 static features designed to capture structural properties of MIP formulations, building on an earlier feature set introduced by Hutter et al. (2013). We use this feature set as a baseline to contextualize the representational efficiency of FORGE. Let us note that these hand-crafted approaches have been engineered carefully over the years and remain solver-dependent.

Table 1 enumerates a compressed set of static features from Hutter et al. (2014). As shown in Figure 8, static features performs remarkable well. Likewise, FORGE, which operates on only 10 input features (6 per variable and 4 per constraint), achieves good clustering performance in terms of NMI relative to the full 100-feature baseline. This result suggests that the learned representations produced by FORGE capture sufficient structural information to rival a feature set that is an order of magnitude larger, carefully curated over multiple iterations of prior work, and most importantly, remain solver-dependent.

When we compare static vs. learned methods from a time-to-compute perspective, FORGE offers a computational advantage. When limited to a single process, extracting the 100 static features across all instances requires approximately 900 seconds, whereas FORGE completes inference in approximately 200 seconds. Taken together, our general-purpose, instance-level FORGE embeddings delivers an efficient and compact alternative to the traditional hand-engineered MIP feature sets.

| Feature family (pattern) | Level | Brief definition |
|---|---|---|
| `probtype, n_vars, n_constr, n_nzcnt, nq_vars, nq_constr, nq_nzcnt` | Instance | Problem type code and size metadata (linear/quadratic counts). |
| `num_{b,i,c,s,n}_variables, ratio_{b,i,c,s,n}_variables` | Instance | Counts and ratios by variable type ($B, I, C, S, N$). |
| `num_i+_variables, ratio_i+_variables` | Instance | Count/ratio of non-continuous variables (VType $\neq C$). |

| Feature family (pattern) | Level | Brief definition |
|---|---|---|
| `num_unbounded_disc`, `ratio_unbounded_disc` | Instance | Count/ratio of unbounded integer-like vars ($I$ or $N$ with an infinite bound). |
| `support_size_{avg,median,varcoef,q90mq10}` | Instance | Stats over per-variable support size (for bounded non-continuous vars): 2 for $B, S$, $ub - lb + 1$ for bounded $I$, $ub - lb + 2$ for bounded $N$. |
| `rhs_c{c}_{avg,varcoef}` | Instance | RHS stats within each constraint-sense bucket $c$. |
| `vcg_constr_deg{s}_{stat_deg}` | Instance | Stats of row nonzero counts in $A_{:,S_s}$ (constraint degree wrt var set $s$). |
| `vcg_var_deg{s}_{stat_deg}` | Instance | Stats of column nonzero counts in $A_{:,S_s}$ (variable degree wrt var set $s$). |
| `vcg_constr_weight{s}_{stat_w}` | Instance | Stats of row sums in $A_{:,S_s}$. |
| `vcg_var_weight{s}_{stat_w}` | Instance | Stats of column sums in $A_{:,S_s}$. |
| `A_ij_normalized{s}_{stat_w}` | Instance | Stats of flattened values $a_{ij}/rhs_i$ for rows with $|rhs_i| > 10^{-6}$, restricted to set $s$. |
| `a_normalized_varcoefs{s}_{stat_w}` | Instance | For each row: normalize absolute row coefficients to sum 1, compute varcoef; then aggregate those row-wise values. |
| `obj_coefs{s}_{stat_obj}` | Instance | Stats of $|c_j|$ over active vars in set $s$. |
| `obj_coef_per_constr{s}_{stat_obj}` | Instance | Stats of $|c_j|/\deg_s(j)$ over constrained vars ($\deg_s(j) > 0$). |
| `obj_coef_per_sqr_constr{s}_{stat_obj}` | Instance | Stats of $|c_j|/\sqrt{\deg_s(j)}$ over constrained vars ($\deg_s(j) > 0$). |
| `time_VCG{s}` | Instance | Runtime (seconds) to compute graph-family features for set $s$. |
| `is_type_{B,I,C,S,N}`, `is_discrete`, `is_unbounded_discrete`, `support_size`, `obj_abs` | Variable | Per-variable type indicators and scalar attributes. |
| `vcg_var_deg{s}`, `vcg_var_weight{s}`, `obj_coef_per_constr{s}`, `obj_coef_per_sqr_constr{s}` | Variable | Per-variable structural features for each set $s$. |
| `rhs, sense_le, sense_eq, sense_ge` | Constraint | Per-constraint RHS and one-hot sense indicators. |
| `vcg_constr_deg{s}`, `vcg_constr_weight{s}`, `a_normalized_varcoefs{s}` | Constraint | Per-constraint structural features for each set $s$. |

Table 1: Features adapted from Hutter et al. (2014)

Notation used for the table:

$s \in \{0, 1, 2\}, \quad s = 0:$ non-continuous vars (VType $\neq C$), $s = 1:$ continuous vars (VType $= C$), $s = 2:$ all vars

$c \in \{0, 1, 2\}, \quad c = 0:$ sense $<$, $c = 1:$ sense $=$, $c = 2:$ sense $>$

$\text{stat}_{\text{deg}} \in \{\text{avg}, \text{median}, \text{varcoef}, \text{q90mq10}\}$

$\text{stat}_{\text{w}} \in \{\text{avg}, \text{varcoef}\}, \quad \text{stat}_{\text{obj}} \in \{\text{avg}, \text{std}\}$

## A.2 Additional Clustering Results on strIPlib

To further validate our clustering results from §4, where we pre-trained FORGE on 1,800 MIPLIB instances and clustered instances from D-MIPLIB, we repeat the experiment using our MIPLIB-pre-trained FORGE to cluster instances from strIPlib (Bastubbe et al., 2025). We select 50 instances from each of 10 previously

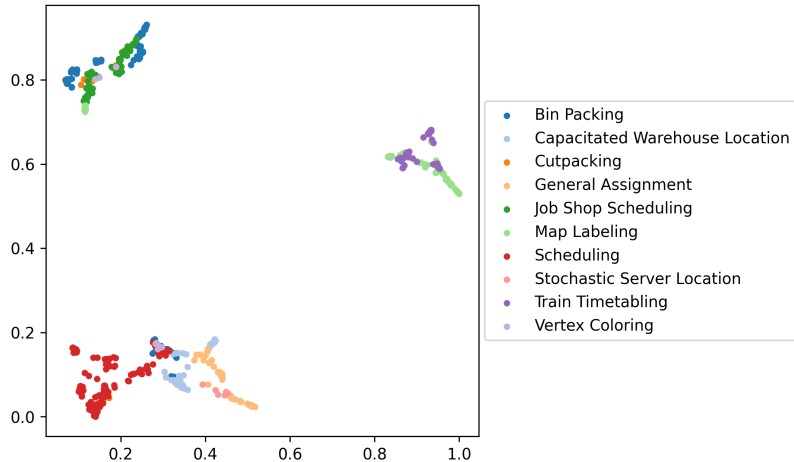

Figure 9: Visualization of MIP embeddings from strIPlib. FORGE has never seen these instances, was only trained on 1,800 MIPLIB instances, yet still, manages to cleanly cluster strIPlib problems.

unseen problem types: Bin Packing, Capacitated Warehouse Location, Cutpacking, General Assignment, Job Shop Scheduling, Map Labeling, Scheduling, Server Location, Train Timetabling, and Vertex Coloring.

Figure 9 shows 2D clustering visualizations using PaCMAP. As in our previous study, these strIPlib instances were unseen by FORGE, yet the model still clusters different problems cleanly. For example, all packing instances group together in the top-left, while warehouse and server location instances cluster in the bottom-left. Other interesting patterns emerge, such as Train Timetabling and Map Labeling appearing close to each other, suggesting potential transfer learning opportunity.

## A.3 Ablation Study on The Codebook Size

An important consideration in vector quantization is determining the appropriate codebook size to effectively capture global structural patterns. While increasing the number of codes might intuitively enhance representation capacity, empirical evidence shows this relationship is non-monotonic due to *code under-utilization*, where some codes remain unused during training. This phenomenon has been extensively studied in domains such as speech and computer vision (Yu et al., 2022; Zeghidour et al., 2021; Lee et al., 2022). To examine this trade-off, we conduct an ablation study measuring NMI scores for clustering across 1,050 D-MIPLIB instances. Following the setup in §4, we generate FORGE embeddings with varying codebook sizes.

Table 2 reports our results showing no statistically significant difference in clustering performance, measured by NMI against the ground truth, across different codebook configurations. We attribute this stability to the clustering objective, which appears less sensitive to codebook size variations compared to downstream predictive tasks. This observation aligns with Yang et al. (2024), who reported consistent classification accuracy across various graph datasets until codebook sizes exceeded 16,000, at which point performance began to degrade. Based on these findings, we set the codebook size to 5,000 in our experiments (§5.1 and §5.2). While our pre-trained FORGE model was trained on 2,850 MIP instances, making 5,000 codes potentially excessive, the framework is designed as a foundational architecture ready for extension to much larger datasets.

Table 2: Ablation study on codebook size of the FORGE architecture measuring NMI scores.

| Codebook Size | NMI |
|---|---|
| 500 | $0.810 \pm 0.027$ |
| 1,000 | $0.818 \pm 0.030$ |
| 2,500 | $0.822 \pm 0.026$ |
| **5,000** | $\mathbf{0.843 \pm 0.031}$ |
| 10,000 | $0.805 \pm 0.022$ |

### A.4 Vector Arithmetic in the Latent MIP Embedding Space

Given MIP embeddings across various problem types, we ask if we can identify certain 'directions' in this latent optimization embedding space that could shift a MIP instance from one problem type to another based on our theoretical understanding. This line of reasoning is inspired by the earlier works on understanding analogies in word embeddings, as in the famous $King - Man + Woman \approx Queen$ example (Ethayarajh et al., 2018). Concretely, we inspect the relationship among the following *covering* and *packing* problems:

**Set Cover Problem (SCP):** Given a universe of elements and a collection of subsets, find the smallest number of subsets to cover all elements.

$$min \sum_{S \in \mathcal{S}} x_S$$

$$\sum_{S:e \in S} x_S \geq 1 \quad \forall e \in \mathcal{U}$$

$$x_S \in \{0,1\} \quad \forall S \in \mathcal{S}$$

**Vertex Cover Problem (VCP):** Given a graph, find the smallest set of vertices such that every edge has at least one endpoint in the set.

$$min \sum_{v \in V} x_v$$

$$x_u + x_v \geq 1 \quad \forall (u,v) \in E$$

$$x_v \in \{0,1\} \quad \forall v \in V$$

**Bin Packing Problem (BPP):** Given a set of items and a collection of bins with a capacity, find the smallest number of bins that pack all items within bin capacities.

$$min \sum_{j=1}^{n} y_j$$

$$\sum_{j=1}^{n} x_{ij} = 1 \quad \forall i \in \{1, \ldots, m\}$$

$$\sum_{i=1}^{m} w_i x_{ij} \leq C \cdot y_j \quad \forall j \in \{1, \ldots, n\}$$

$$x_{ij} \in \{0,1\}, \quad y_j \in \{0,1\}$$

**Independent Set Problem (ISP):** Given a graph, find the largest set of vertices such that no two selected nodes are adjacent.

$$max \sum_{v \in V} x_v$$

$$x_u + x_v \leq 1 \quad \forall (u,v) \in E$$

$$x_v \in \{0,1\} \quad \forall v \in V$$

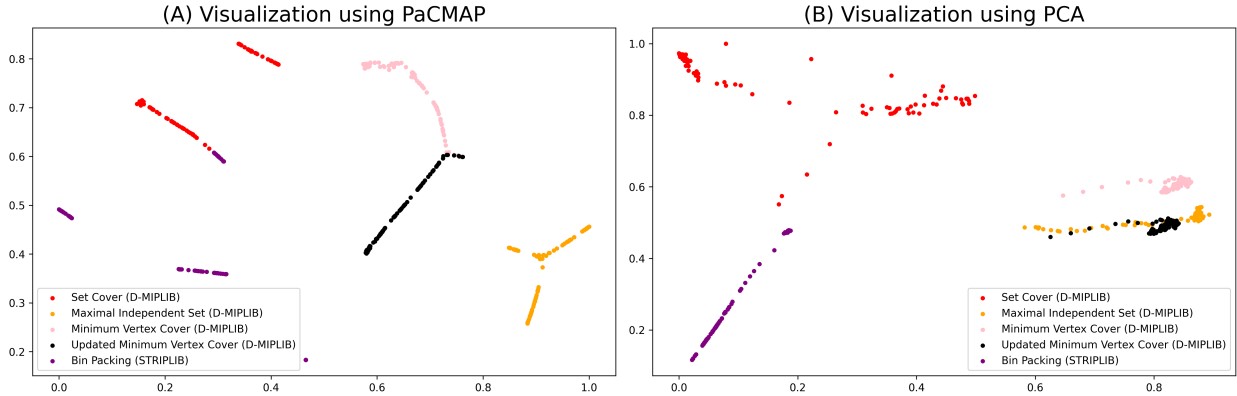

Figure 10: A (stretch) analogy of $King - Men + Woman \approx Queen$ for Combinatorial Optimization: $Vertex\ Cover - Set\ Cover + Bin\ Packing \approx Independent\ Set$. The updated Vertex Cover instances are shown in **black** which move closer to the Maximal Independent Set instances.

**Observation:** In these problem definitions and formulations, notice how Set Cover and Vertex Cover are similar to each other as *covering* problems, and Bin Packing and Independent Set are similar to each other as *packing* problems. Conversely, the pairs are complementary to each other, e.g, the sum of minimum vertex cover and maximum independent set equals to the number of vertices.

**Research Question:** We ask the following question: if we identify the difference in dimensions between *covering* and *packing*, and then push Minimum Vertex Cover instances along that direction, do we obtain embeddings that are closer to Maximal Independent Set instances? The intuition behind this is to remove the 'cover' aspect from Vertex Cover, and then add to that the 'packing' aspect of Bin Packing, to obtain Independent Set as a result.

To validate this experimentally, we fix the graph size to 1,000 vertices and select 50 random instances each for Set Cover, Vertex Cover, Independent Set from D-MIPLIB (Huang et al., 2024), and 50 random Bin Packing from strIPlib (Bastubbe et al., 2025). We also control for the problem difficulty by ensuring all instances are solvable by GUROBI within 60 seconds.

**Vector Arithmetic for Optimization:** Next, we apply vector arithmetic on the embeddings of MIP instances to verify our intuition, as a (stretch) analogy to the famous $King - Man + Woman \approx Queen$ example, and test for $VertexCover - SetCover + BinPacking \approx IndependentSet$. This analogy is not perfect, as there is no *single* 'King' instance in the optimization embedding space, but instead, we average over multiple instances while controlling for the same graph size and similar difficulty across problems.

Given the embeddings of instances for Set Cover, $e_{sc}$, Set Packing, $e_{bp}$, Vertex Cover, $e_{mvc}$, and Independent Set, $e_{mis}$, where $e \in \mathbb{R}^{|instances| \times |codes|}$, we compute the difference in dimensions between covering and packing, $d_{sc-bp}$, as follows:

$$\mu_{sc} = mean(e_{sc}, axis = 0) \in \mathbb{R}^{1 \times |codes|} \tag{6}$$

$$\mu_{bp} = mean(e_{bp}, axis = 0) \in \mathbb{R}^{1 \times |codes|} \tag{7}$$

$$d_{sc-bp} = \mu_{sc} - \mu_{bp} \in \mathbb{R}^{1 \times |codes|} \tag{8}$$

Given the difference between covering and packing, $d_{sc-bp}$, we update the embedding of the instances of minimum vertex cover problem:

$$e_{updated\_mvc} = e_{mvc} - d_{sc-bp} \in \mathbb{R}^{|instances| \times |codes|} \tag{9}$$

**Results:** Figure 10 visualizes these vector operations in the latent MIP embedding space using PaCMAP in Figure 10-A and PCA in Figure 10-B. In both visualizations, notice how the embeddings of **updated** Vertex Cover move closer to the embeddings of Independent Set compared to their initial starting point, once they are modified by the direction obtained from the difference of embeddings of covering and packing instances.

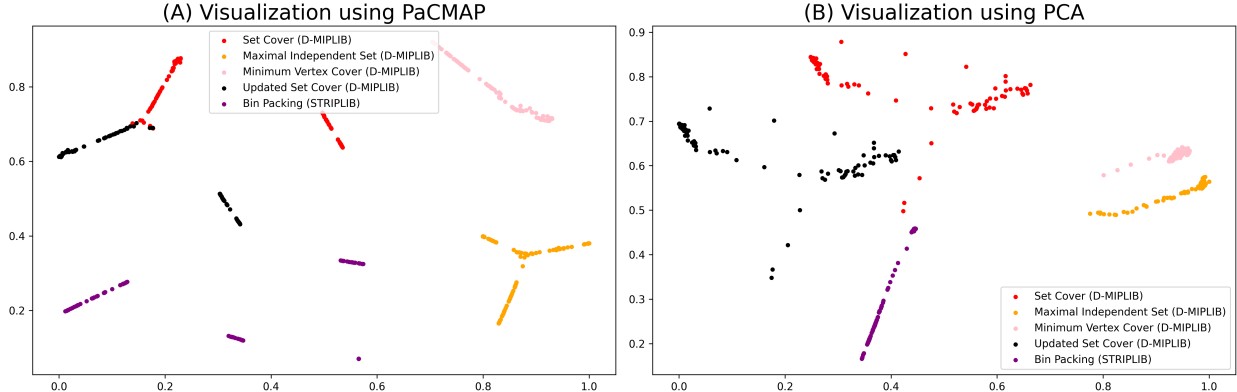

Figure 11: A (stretch) analogy of $King - Man + Woman \approx Queen$ for Combinatorial Optimization: $Set\ Cover - Vertex\ Cover + Independent\ Set \approx Bin\ Packing$. The updated Set Cover instances are shown in **black** that move closer to the Bin Packing instances.

Similarly, we further examine the vector arithmetic results for:

- In Figure 11 above, $Set\ Cover - Vertex\ Cover + Independent\ Set = Bin\ Packing$: Intuitively, this is can be understood as removing the 'cover' direction from Set Cover by subtracting Vertex Cover and adding the 'packing' direction from Independent Set to push the **updated** Set Cover instances closer to the Bin Packing instances from the initial Set Cover instances.

- In Figure 12 below, $Independent\ Set - Set\ Packing + Set\ Cover = Vertex\ Cover$: Intuitively, this is can be understood as removing the 'packing' direction from Independent Set by subtracting from Bin Packing and adding the 'cover' direction from Set Cover to push the **updated** Independent Set instances closer to the Vertex Cover instances from the initial Independent Set instances.

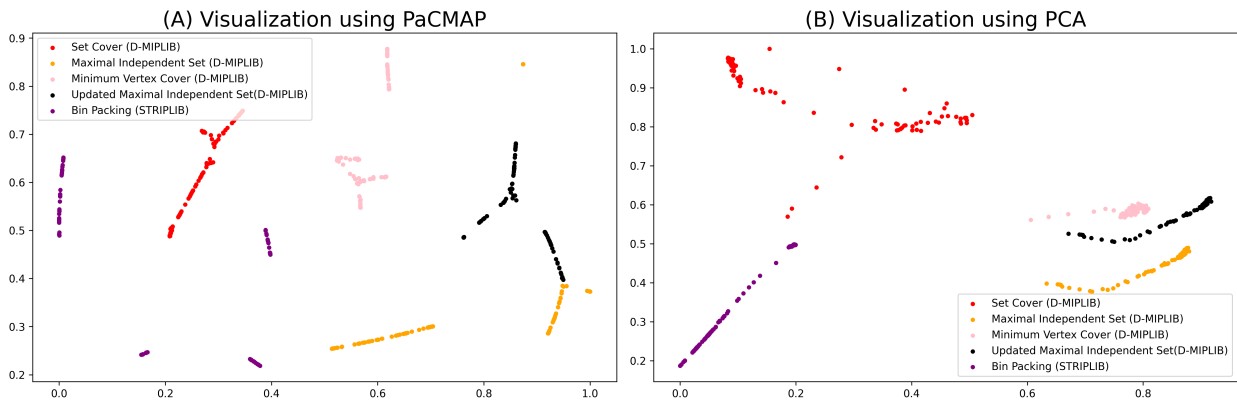

Figure 12: A (stretch) analogy of $King - Men + Woman \approx Queen$ for Combinatorial Optimization: $Independent\ Set - Bin\ Packing + Set\ Cover \approx Vertex\ Cover$. The updated Independent Set instances are shown in **black** that move closer to the Minimum Vertex Cover instances.

## A.5 Additional Downstream Task: Solver Configuration

To further evaluate the generalization of FORGE to other optimization tasks beyond integrality gap prediction (Section 5.1) and and search guidance (Section 5.2), we perform a preliminary study on an additional downstream optimization tasks as recently studied in Cai et al. (2025a), Solver Configuration Prediction.

| Problem | Metric | Multi-task Training (Cai et al., 2025a) | Fine-tuned FORGE |
|---------|--------|------------------------------------------|------------------|
| CA | Mean Runtime(s) | 801.21 | 670.95 |
| MVC | Mean Runtime(s) | 819.87 | 735.79 |
| IS | Mean Runtime(s) | 718.28 | 566.86 |

Table 3: Computational efficiency on 50 CA, MVC, and IS test instances following the benchmark in Cai et al. (2025a) for Configuration Prediction using SCIP as the backend solver.

Solver configuration is aimed at finding the best hyper-parameters for the options exposed in the solver to best fit the instance at hand. To ensure a fair comparison with Cai et al. (2025a), we adopt the same dataset, their multitask training pipeline as established by Cai et al. (2025a) [3]. We retrain their multitask model from scratch on their Combinatorial Auction (CA), Minimum Vertex Cover (MVC), and Independent Set (IS) problem instances. To fine-tune FORGE on the new solver configuration prediction task, we introduce task-specific prediction heads, similar those described in Sections 5.1 and 5.2 and illustrated in Figure 4.

It is important to note that, the multi-task learning approach of Cai et al. (2025a) is trained *per problem domain* using two task heads *jointly*. This yields a dedicated model per problem. In contrasts, we start with our pretrained Forge and fine-tune it on *all problems* for each task. This highlights a complementary axis of generalization: while Cai et al. (2025a) shares an architecture across tasks but specializes per problem, FORGE shares an architecture across problems but specializes per task. A natural direction for future work is to unify both axes, i.e., training a single model that jointly addresses all tasks across all problem types.

Following the same setup in Cai et al. (2025a), Table 3 presents results for configuration prediction using SCIP as the backend solver. We confirm that our results from re-running Cai et al. (2025a) reproduces similar trends, apart from hardware differences yielding minor discrepancy in exact runtimes in seconds.

In Table 3, the results are comparable to each other. Let us again highlight the complementary nature of this two works. It is remarkable that Cai et al. (2025a) enables training for *both of these tasks in a single network*, and similarly, FORGE enables fine-tuning across *all problems at once.*

### A.6 Additional Results on Transfer from Optimization to Satisfaction

As a further analysis of FORGE on *other combinatorial problems*, we evaluate it on Boolean Satisfiability (SAT) instances.[4] We use instances from G4SATBench (Li et al.), a widely used benchmark for GNN-based SAT reasoning. The benchmark contains seven datasets drawn from three domains: random, pseudo-industrial, and combinatorial SAT problems. Each dataset provides easy, medium, and hard difficulty levels with dedicated train and test splits, and each split is balanced between SAT and UNSAT instances. This diversity makes G4SATBench a suitable testbed for evaluating cross-domain generalization of SAT embeddings. In our clustering experiments, we use all test instances from the Hard category, yielding 1400 unseen formulas: 200 from each of the seven problem types, with an equal split of 100 SAT and 100 UNSAT instances. This results in 14 ground-truth groups defined by (problem type, feasibility) pairs. We focus on the Hard split for clarity, as results on the Easy and Medium categories follow similar trends.

In terms of FORGE models, we use the following:

- FORGE-MIP (zero-change transfer): This variant directly applies our pretrained FORGE model to SAT instances encoded as MIPs. Both the architecture and the MIP-specific node features are kept **unchanged**. This represents pure weight-level transfer from optimization to satisfiability.

---

[3] We are immensely grateful to our authors for making their pipeline available in open-source that enabled this fair comparison.
[4] We would like to thank Koyena Pal for her help running these experiments.

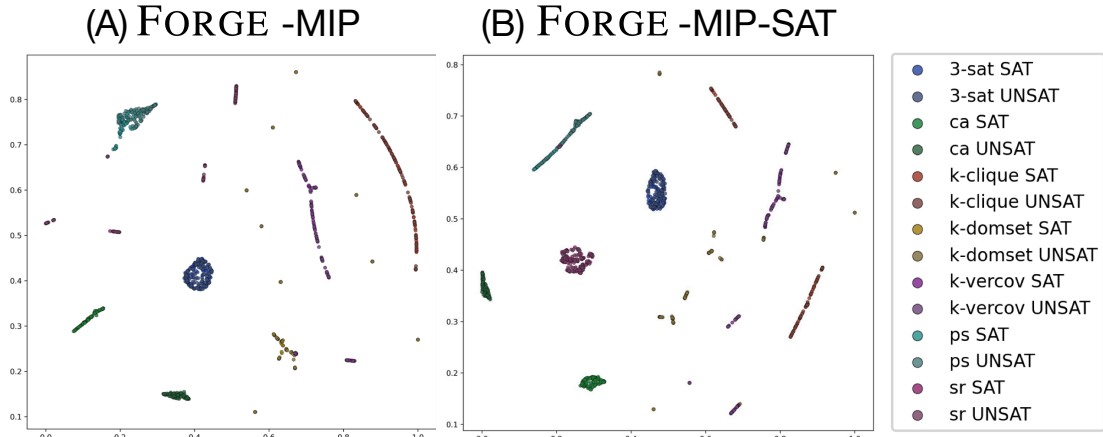

Figure 13: Instance level embeddings of SAT instances projected in 2D using PaCMAP. Forge-MIP and Forge-MIP-SAT both yield surprising results showing well separated clusters despite being trained on the MIPLIB dataset.

- FORGE-MIP-SAT (feature-adapted transfer): This variant also reuses our pretrained MIP weights but replaces the MIP node features with SAT-specific features. The graph structure and pretrained weights remain unchanged, isolating the effect of feature specialization.

SAT Node Features: For FORGE-MIP-SAT, we use SAT-specific constraint node features described by clause-level properties: width (number of literals in the clause), pos_count (number of positive literals), neg_count (number of negative literals), and pos_neg_ratio (ratio of positive to negative literals). SAT-specific variable node features are described by degree-based properties: degree (total number of clauses the variable appears in), pos_deg (number of clauses where it appears positively), neg_deg (number of clauses where it appears negated), pos_neg_ratio (ratio of positive to negative appearances), pos_deg_norm (positive degree normalized by the mean positive degree across all variables), and neg_deg_norm (negative degree normalized by the mean negative degree across all variables).

Figure 13 shows the instance-level embeddings projected into 2D using PaCMAP. In the ideal scenario, the embeddings would form 14 distinct clusters accordingly with seven problems and their SAT/UNSAT labels. For NMI, a perfect score is 1.0, whereas a uniform random assignment over the 14 ground-truth groups would yield roughly $1/14 \approx 0.07$. Both FORGE-MIP and FORGE-MIP-SAT yield good results showing well separated clusters. Despite being trained only on MIP datasets they significantly improve the random control NMI $approx 0.07$ with to $0.76 \pm 0.001$ and $0.77 \pm 0.003$ respectively. **It is remarkable that embeddings trained on MIP datasets transfer so effectively to the G4SATBench dataset.**

Our initial results, detailed further in (Pal & Kadıoğlu, 2026), on transfer learning from MIP to SAT opens an exciting directions for future research: in principle, it is now possible to train a FORGE model on a *mixture of MIP and SAT instances, enabling hybrid, multi-domain pretraining.* This is a step toward a foundational model that unify optimization and satisfaction and support both supervised and unsupervised tasks.

### A.7 Details of Our Experimental Setup

For the experiments presented in §5, we train on the ml.g5.xlarge AWS instance with a GPU with 24 GB of memory. Inference experiments were run on the ml.c5.12xlarge instance with 48 cores and 96 GB of RAM. To ensure consistency and fairness, all experiments were executed with GUROBI (v12.0.3), a state-of-the-art commercial MIP solver (Gurobi Optimization, LLC, 2024), restricted to a single thread and a time limit of 3600 seconds. Unsupervised pre-training and integrality gap fine-tuning are run for 10 epochs with a learning rate of $10^{-4}$, while the search guidance prediction task is trained for 25 epochs using a learning rate of $10^{-5}$. The fixed radius in search guidance is set to 0.1.

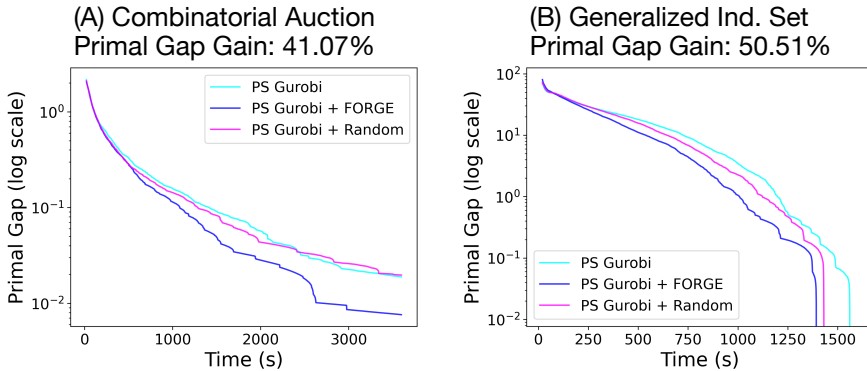

Figure 14: PS-Gurobi vs. PS-Gurobi + Forge vs. PS-Gurobi + Random Search Guidance. Each subplot shows primal gap (the lower, the better) averaged across 50 medium instances in each problem.

## A.8 Ablation Study on Search Guidance

In §5.2, as part of our comparison with state-of-the-art ML methods, we augmented PS-Gurobi (Han et al., 2023) with Forge embeddings and observed performance gains from their combination. More precisely, in PS-Gurobi, the input node features for constraints and variables are 4 and 6 dimensions, respectively. To minimize architectural changes, we applied Principal Component Analysis (PCA) to reduce the dimensionality of Forge embeddings from 1024 down to 64. This raises an important question: how much of the improvement is due to Forge versus the increased model capacity from higher-dimensional inputs? To control for this effect, we repeated the experiments from §5.2, augmenting PS-Gurobi with 64-dimensional random vectors. This comparison isolates the contribution of our pre-trained embeddings to overall performance. Each PS-Gurobi model for each problem type and augmentation was trained from scratch.

Figure 14 compares PS-Gurobi, PS-Gurobi augmented with Forge, and PS-Gurobi with random vectors. Our ablation shows that adding random vectors occasionally improves performance, likely due to the increase in input dimensionality, from 4 and 6 features for constraints and variables to 68 and 70, resulting in greater model capacity. However, semantic embeddings from Forge consistently dominate, delivering the strongest performance across both problem types.

## A.9 Additional Related Work

Recent advances in applying machine learning to combinatorial optimization, particularly Mixed-Integer Programming (MIP), have led to a wealth of literature. These methods can be broadly categorized based on their focus areas, such as solver configuration, branching strategies, heuristic design, and generalization across problem types.

**Theoretical foundations:** A stream of of works by Chen et al. systematically establishes what GNNs can provably represent across the hierarchy of linear Chen et al. (2022) and mixed-integer linear Chen et al. (2023a) programs. First, Chen et al. (2022) proved that message-passing GNNs operating on a bipartite variable-constraint graph can uniformly approximate the feasibility, optimal objective value, and optimal solution mappings of LPs. Then, Chen et al. (2023a) extended this analysis to mixed-integer linear programs, introducing the notion of *foldable* MILPs. Foldable instances have variable–constraint symmetries that cause the Weisfeiler–Lehman (WL) test to assign identical colors to distinct vertices, making them indistinguishable to any GNN. They prove that GNNs achieve uniform approximation guarantees on *unfoldable* instances, while appending random features breaks foldability with probability ones. Forge reinforcements these theoretical foundations with scalable pre-trained model with practical improvements that goes beyond meaningful representation to improving MIP solvers. Extending the theoretical framework from GNN architecture for MIP representation to a theory behind the foundational Forge architecture is a valuable future direction.

**Learning-based Methods to Enhance MIP Solvers:** Several works have explored integrating supervised and reinforcement learning into traditional MIP solving pipelines. The survey by Zhang et al. (2023) categorizes these methods into two main groups: those enhancing the Branch-and-Bound process (e.g., branching variable prediction (Kadıoğlu et al., 2012; Liberto et al., 2016; He et al., 2014b; Lodi & Zarpellon, 2017; Bengio et al., 2021a), cutting plane selection (Tang et al., 2020; Paulus et al., 2022), node selection He et al. (2014a)), and those improving heuristic algorithms such as Large Neighborhood Search, Feasibility Pump, and Predict-and-Pick. These models are typically trained end-to-end, with reinforcement learning often relying on imitation learning. We refer to (Gast Zepeda et al., 2025) for recent survey.

In the context of solver configuration, the earlier work of Kadıoğlu et al. (2010) propose the ISAC framework that takes a clustering-based approach to instance-specific algorithm configuration. It uses g-means to cluster problem instances and assigns configurations based on cluster membership. The method considers domain-specific features such as cost-density ratios and root cost metrics for problems such as SCP, MIP, and SAT. Algorithm selection, scheduling and portfolios have also been studied (Xu et al., 2008; Kadıoğlu et al., 2011a; Schede et al., 2025; Kemminer et al., 2024). More recently, Hosny & Reda (2024) propose predicting solver parameters by leveraging similarities between problem instances. Their key assumption is that instances with similar costs under one configuration will behave similarly under other configurations. Their features include pre-solve statistics and tree-based metrics, with a triplet loss guiding the learning process using solved instance objectives.

To improve generalization, Boisvert et al. (2024) propose a generic representation for combinatorial problems using abstract syntax trees with five node types - variables, constraints, values, operators and a model node. While expressive, this approach results in large, computationally expensive graphs.

**Multi-Task and Generalist Models:** Similar to our work here, efforts to unify learning across tasks and problem types have also emerged. Cai et al. (2025a) introduce a multi-task representation learning framework for MIP, training a shared backbone across tasks such as backdoor prediction and solver configuration prediction, followed by fine-tuning for specific problem types. Their method uses a bipartite graph representation, Graph Attention Networks (GAT), and contrastive loss, and is evaluated on problems such as CA, MVC, and (MIS). Similarly, Drakulic et al. (2024) present GOAL, a generalist agent for combinatorial optimization that learns to solve diverse combinatorial optimization problems through imitation learning. Unlike FORGE, it requires supervised training via problem-specific adapters. It avoids GNNs, instead using mixed attention over edge and node matrices derived from bipartite graphs. Li et al. (2025) propose an LLM-based evolutionary framework that can generate a large set of diverse MIP classes and can be fine tuned to predict integrality gaps and branching nodes.

**Graph Neural Networks for Branching and Heuristics:** Graph-based representations have become standard for encoding MIP instances. One of the earliest work on graph-based learning by Gasse et al. (2019) uses GNNs to learn strong branching policies, introducing a bipartite graph structure and dual half-convolutions for message passing between constraints and variables. Chen et al. (2024) revisit GNN for MIPs and show that higher-order GNNs can overcome limitations identified via the 1-Weisfeiler-Lehman test, making all instances tractable for message passing. Cantürk et al. (2024) introduce improvements to the standard GNN workflow for CO so that they generalize on instances of a larger scale than those used in training and propose a two-stage primal heuristic strategy based on uncertainty quantification to automatically configure how solution search relies on the predicted decision values.

Along the lines of Backdoor learning, Cai et al. (2024a) use Monte Carlo Tree Search to identify effective backdoors, training a GAT to score variables. Ferber et al. (2022) propose pseudo-backdoors, using one model that characterizes if a subset of variables is a good backdoor and another model to predict whether prioritizing this subset would lead to a smaller run time.

**Learning Heuristics and Large Neighborhood Search (LNS):** Earlier works such as (Kadıoğlu et al., 2017; 2011b) proposes learning reactive restart and impact-based strategies to improve search. Recently, a growing body of work focuses on learning heuristics, particularly for Large Neighborhood Search (LNS). Huang et al. (2023) use expert heuristics to create training data followed by random perturbations to create 'negative' samples. Then contrastive learning is used to train GATs to predict node probabilities. Other

works by Wu et al. (2021) and Song et al. (2020) use deep reinforcement learning to learn destroy operators or decompositions, with rewards based on objective improvements. Khalil et al. (2017) model the success of heuristics at specific nodes by examining instance based characteristics and use logistic regression over a rich feature set, including LP relaxation and scoring metrics. Cai et al. (2025b) propose BALANS, an adaptive meta-solver for MIPs with online learning capability that does not require any supervision or apriori training. During the search, the selection among different neighborhood definitions is guided on the fly via multi-armed bandit algorithms (Strong et al., 2021; 2019).Yilmaz et al. (2025) extend BALANS Cai et al. (2024b) using solver- and algorithmic-level parallelism into PARBALANS to improve performance on challenging MIP instances.

**Problem Specific Methods:** The vehicle routing problem (VRP) has garnered special attention from the community (Berto et al., 2025; Hottung & Tierney, 2022).Zhou et al. (2023) introduce a meta-learning framework for VRPs, enabling generalization across problem sizes and distributions. Berto et al. (2024) explore ML solutions for different kinds of VRPs like those including backhauls, multi-depots, duration limits, mixed backhaul, line hauls, among others. They use a common encoder for all VRP types with global attributes for problem type and local node attributes to capture customer specific attributes such as location and demands.

Another problem that has garnered attention is the constraint satisfaction problem. Tönshoff et al. (2023) use GNNs to predict soft assignments, with reinforcement learning rewards based on constraint satisfaction improvements. Duan et al. (2022) propose a contrastive learning framework that generates label-preserving augmentations for SAT problems. These include techniques such as unit clause propagation, pure literal elimination, and clause resolution, ensuring that the satisfiability of the instance remains unchanged while enhancing the model's robustness. Shafi et al. (2025) introduce Graph-SCP, a method that leverages features extracted from both bipartite and hypergraph representations of SCP instances. A GNN is then trained with these features to predict a promising subproblem where the optimal solution is likely to reside. This predicted subproblem is passed to a solver, effectively accelerating the overall solution process.

**Unsupervised Approaches:** Unsupervised learning has also been explored in various forms. Karalias & Loukas (2020) introduce a framework that learns a probability distribution over nodes, optimizing a loss that bounds the probability of finding a solution. These are then decoded using a derandomization process. Bu et al. (2024) build on the work by Karalias & Loukas (2020) by formalizing objective construction and derandomization strategies. They derive explicit formulations tailored to a range of combinatorial problems, including facility location, maximum coverage, and robust graph coloring. Sanokowski et al. (2024) provide an approach for solving combinatorial optimization problems without labeled data by leveraging diffusion models to sample from complex discrete distributions. Their method avoids the need for exact likelihoods by optimizing a loss that upper bounds the reverse KL divergence. While FORGE falls in the domain of unsupervised approaches, we differ in that our goal is to not solve a given instance in an unsupervised manner, but rather to learn the graphical structure of various MIP instances in an unsupervised manner followed by supervised fine-tuning to aid in finding the solution.

