# OpenReview forum: "Forge: Foundational Optimization Representations from Graph Embeddings"
_TMLR — Accepted by TMLR_

### Review · Reviewer_h6nc · 2026-04-05

**Summary Of Contributions:**

This paper introduces Forge, a representation learning framework for mixed-integer programming (MIP) based on a vector-quantized graph autoencoder. The method works on the bipartite graph representation of an MIP instance, using static variable/constraint features for unsupervised pretraining, and then derives an instance-level embedding from the distribution of learned discrete codes. The paper evaluates these representations on two downstream tasks: integrality-gap prediction for pseudo-cut generation, and variable-level search guidance for solver assistance. The paper addresses an important and interesting idea of learning reusable MIP representations without relying on optimal solutions. And the use of vector quantization to construct discrete optimization vocabularies is a distinctive design. However, this paper makes very broad claims about generalization and “foundational” representations, but the empirical evidence appears weaker than the claims.

**Audience:**

Yes

**Audience Explanation:**

The idea of learning unsupervised or foundational representations for MIP instances is relevant to readers' interest in representation learning, graph-based learning, and ML for optimization.

**Claims And Evidence:**

No

**Claims Explanation:**

1. Overclaiming the framing of Forge as a foundational model with broad generalization. The pretrained model is compact, while the downstream validation is restricted to two tasks and a fairly narrow set of evaluations. Lack of interpretability limits both the trustworthiness and the practical value of the work.

2. SOTA comparisons not fully convincing as currently presented. Some are not under the same experimental protocol or dataset (5.2), and part of the gain in the PS-Gurobi augmentation may come from increased input dimensionality/model capacity(Appendix A.5)

3. While this remains a critical central issue, the writing about the training and testing data remains unclear, which makes it harder to assess the potential data contamination problem.

**Requested Changes:**

1. (Critical) Soften the claim of “foundational model” and the scope of the generalization, unless additional evidence is added.
2. (Relatively critical) Compared with the baseline in fair settings.
3. (Relatively critical) Add interpretability analysis of the learned codebook/vocabulary
4. (Recommended) Improve the presentation of the datasets for unsupervised pretraining, downstream fine-tuning, and final testing to resolve ambiguities

---

> ### Author Response · Authors · 2026-04-25
> **“(Critical) Soften the claim of “foundational model” and the scope of the generalization, unless additional evidence is added.”**
>
> Thank you for your constructive feedback. **We shall certainly soften our claims**. In particular, Forge must be treated “as a step toward a foundational model” and not “the foundational model”. We appreciate this remark and accordingly reworded the Introduction (changes are in blue for ease of your evaluation)
>
> With that, let us highlight:
>
> - The key issue with current ML methods for CO is 1) their supervised nature and dependency on labels/solvers and 2) requirement to train one model for each problem type. Forge is an attempt to address both.
>
> - There is recent emerging work for supervised training for multiple downstream tasks (Cai et. al. 2025), but again, this is trained specific on each problem. Going beyond multi-task learning, our primary contribution pretraining for multiple tasks aimed across problems. This gives us the ability to use a single pretrained model with the same weights across multiple CO problem types and on two very different tasks.
>
> - Let us share a historical perspective: In 2022, the Dagstuhl Seminar on Data-Driven Combinatorial Optimization convened a working group explicitly around the concept of a Foundation Model for CO (please see page 3, https://tinyurl.com/4fm8medn). Four years later, there still exists no general-purpose, unsupervised MIP embedding despite remarkable advances in parallel domains such as NLP & CV. Forge **tries to** address this gap with applicability across problems, sizes, and tasks.
>
>
> We shall further clarify that the term “foundation” is not a reference to scale/size of the models, e.g., when compared to LLMs.
>
> Thank you again, we have edited the introduction to highlight this reframe.

---

> ### Author Response · Authors · 2026-04-25
> **(Relatively critical) Compared with the baseline in fair settings**
>
> PS-Gurobi is an excellent approach that performs well across CO problem types, but with the caveat that it has to be **retrained for each problem type**.
>
> When we compare PS-Gurobi vs. PS-Gurobi+Forge vs. PS-Gurobi + Random Vectors, at each time, we **completely retrain** each PS-Gurobi model from scratch on the dataset corresponding to the problem type (CA vs. IS) so that it’s a fair/best setting for PS-Gurobi.
> The reason we keep these experiments dedicated to two problems (CA & IS) is because PS-Gurobi has hard coded hyperaparameters specific to these problems that we use as is. Please note that, unlike PS-Gurobi, __Forge has never seen this data in its pretraining__.
> We now realize these aspects are not very clear in the paper and have changed this (page 10, highlighted in blue). Thank you!
>
>
> With this setup in place, we further reduce Forge embeddings to a 64 dimensional vector using a linear map (PCA), thereby degrading a large chunk of its predictive power, only to fit into the PS-Gurobi setting.
>
> Despite these, PS-Gurobi + Forge still outperforms PS-Gurobi as well as the vanilla PS-Gurobi + Random to control for comparable complexity.
>
> In Figure 12, we demonstrate that Forge consistently outperforms random augmentation as well, proving that the gains are due to the semantic information in our down-projected embeddings, not merely the increase in input dimensionality.
>
> To your point, PS-Gurobi + Random performs slightly better than PS-Gurobi hinting that there are indeed some gains due to increased capacity but not as much as PS-Gurobi + Forge. As you hinted, that’s why it’s important to keep a control baseline with the same capacity.
>
>
> Notice that in Section 5.2 our main experimental results is Gurobi vs Gurobi + Forge wher we achieve better results than Gurobi on a much harder task. The results with PS-Gurobi is only an additional justification that by simply concatenating Forge embeddings as-is an pre-existing/specialized model (PS-Gurobi), we improve it with minimal effort.
>
> In principle, this opens the door to any existing learning-based method for cut selection, node selection, variable selection to consider concatenation with out-of-the-box pre-trained Forge embeddings.

---

> ### Author Response · Authors · 2026-04-25
> **(Relatively critical) Add interpretability analysis of the learned codebook/vocabulary**
>
> The interpretation of the embeddings is an important/valid direction. This is out of scope of  this initial paper for the first pretrained model that works across tasks and across problems.
>
> That said, in Appendix A.3 we provide one potential interpretability direction using **vector arithmetic in the optimization latent space**, similar to the famous King - Man + Women = Queen analogy.
>
> This experiment provides an initial demonstration of semantic interpretability through latent space that captures fundamental mathematical relationships such as 'Covering' vs. 'Packing' directions. For example, one of our demonstrations in the appendix reflect __Vertex Cover - (Set) Cover + (Bin) Packing ~= Independent Set__
>
> We completely agree with you that this shall be studied further in future.

---

> ### Author Response · Authors · 2026-04-25
> **Improve the presentation of the datasets for unsupervised pretraining**
>
> Thank you for this recommendation.
> We have rewritten sections to make the distinction between train and test splits clearer in page 5, highlighted in blue. We pay special attention to data leakage:
> - In Section 4, Forge is pre-trained on MIPLIB and tested on D-MIPLIB, an entirely different dataset.
> - In Section 5, Forge is pre-trained on MIPLIB + D-MIPLIB (train set) and tested on D-MIPLIB (test set). We can see where the potential confusion arises from and have made this clear in the paper now: D-MIPLIB benchmark cleanly separates train, test and validation splits. We use these splits as-is from the original benchmark similar to previous work that trains and tests on these well-structured splits commonly used in the literature. Thank you for helping us revise this.

---

### Review · Reviewer_Rsii · 2026-04-11

**Summary Of Contributions:**

Learning-based heuristics for branch and bound have gained significant attention: they can improve solver performance on specific problem distributions. Yet these approaches suffer from two limitations: high dependence on solver-generated labels and the need to train separate models from scratch for each problem distribution. FORGE positions itself as a framework to derive general-purpose embeddings for MILP formulations, aiming to address both of these issues. The selling point: there exists a pretrainable model that can produce general-purpose MILP embeddings that, with some fine-tuning, improve solver performance while requiring far fewer labels than training models from scratch. I must concede that this is a strong selling point.

**Audience:**

Yes

**Audience Explanation:**

The paper addresses a recognized gap in the machine learning for combinatorial optimization community: the lack of general-purpose, pretrained embeddings for MILP formulations. Researchers working on learning-based heuristics for branch and bound would benefit from knowing that unsupervised pretraining on diverse MILP instances is viable and can reduce label dependence for downstream tasks. The open-sourced pretrained weights and embeddings add good practical value.

**Broader Impact Concerns:**

No concerns on ethical implications of this work.

**Claims And Evidence:**

No

**Claims Explanation:**

The claim of this paper is directly tied to its selling point: the proposed framework, a vector-quantized graph autoencoder pretrained on diverse MILP instances represented as bipartite graphs, can embed MILP formulations in a way that improves solver performance while requiring less labeled data. The experiments support this claim on only two tasks: integrality-gap prediction for cut generation and variable prediction for search guidance. This is limiting, since some of the most common tasks in learning-based branch and bound are variable, node, and cut selection. As a result, I do not find the presented evidence convincing. Additionally, the baseline comparisons are not consistent: the integrality-gap evaluation uses different test sets from Li et al., which makes it difficult to assess how clearly the evidence supports the claim.

**Requested Changes:**

Please address my concerns above. In particular, I would like to see an experiment on node selection, variable selection, or cut selection (ideally all three) where the pretrained model achieves comparable performance to a bipartite graph model trained from scratch while using significantly less labeled data.

---

> ### Author Response · Authors · 2026-04-25
> **Response to Baseline Consistency in Integrality Gap Evaluation**
>
> We thank the reviewer for noting the difference in test sets.
>
> We are informed by the authors of Li et. al. paper that their exact train/test splits from their paper are **no longer available**. We totally sympathize with this and appreciate their transparent communication with us.
>
> Given the absence of the original data splits used by Li et al., to ensure a rigorous and fair evaluation, we evaluated Forge on ALL of their instances.
>
> This includes __more instances__ and a __more diverse set of instances__ than what’s tested in Li et. al.
>
> Again, Forge has __never seen any of these 17.5K instances__ in its pretraining.
>
> With this setting, we report an error of 18.63% on 17.5K instances across 400 types, compared to Li et al.’s 20.14% on 11.5K instances across 157 types (a subset of what we are testing for).
>
> These results indicate that Forge achieves a lower/competitive error on a larger, more diverse, and out-of-distribution dataset. This provides further evidence of its generalization capabilities.

---

> ### Author Response · Authors · 2026-04-25
> **Response to Experiments on Node, Variable, and Cut Selection**
>
> We agree that node, variable, and cut selection are important tasks in learning-based Branch and Bound (B&B).
>
> Thanks to your remark we have now added **3 net-new experiments** that extend the downstream applicability of Forge as additional evidence.
>
> We include 2 net-new MIP tasks **Backdoor Prediction**  and **Solver Configuration Prediction** with comparison to a recently published work. Moreover, going beyond MIP tasks, we also added a net-new section on Transfer Learning from Optimization to Satisfaction where we use our pretrained MIP model to unsupervised SAT representations to cluster unseen SAT instances.
>
> Please see our commentary to Reviewer uJoj and the new sections at the end of the Appendix.
>
> All in all, we have now showed that our pretrained Forge model can be fine-tuned for important MIP tasks; such as integrality prediction, search guidance, backdoor variable prediction, solver configuration prediction and even transfers from MIPs to SATs.
>
> In more details, these new experiments include:
> 1) Comparison with Cai et. al. CPAIOR’25 Best Paper Award on Multi-Task Learning for CO on
>  1A) **Backdoor Variable Prediction**, and
>  1B) **Solver Configuration Prediction**, and
> 2) **Transfer Learning from Optimization to Decision Problems**
>
> ===
>
> Apart from all these new additional experimental results as you suggested, let us also elaborate on why we selected these two tasks: Task 1 - Integrality Gap Prediction & Task 2 - Search Guidance initially.
>
> Our goal with Forge is to provide a pretrained representation that can be **both** run as on its own (which we show by integrality gap and search guidance) **and** integrated into existing, high-performance learning pipelines rather than replacing them (which we show by integrating with PS-Gurobi).
>
> The pretrained Forge model is specifically designed to provide both local (variable and constraint level) and global (instance-level) embeddings.
>
> In Task 1, we predict the integrality gap of a given instance, which allows us to test **instance-level embeddings.**
>
> We then turn to Task - 2 Search Guidance/Value Hints, which allows us to test **variable-level embeddings.**
>
> Node selection would again use “instance-level” embeddings (which we already test for) and variable selection would guide the search (which we already do via value selection).
>
> Task 1 is a static/zero-shot test (a single cut is added in the beginning from LP prediction and that’s it) vs. Task 2 a dynamic test (value guidance is applied throughout the Bnb).
> This combination allows us to test both root node processing as well as what comes after that. Ideally, one can combine these two together for best performance; add a pseudo-cut from integrality gap prediction at the root note and guide the subsequent search. We are happy to add this combined experimental results if deemed important.
>
> Notice that there exists __no prior work__ that can provide instance level and variable level embeddings with the __same pretrained model__. Above all, this is what we are enabling here.
>
> Moreover, these two tasks allow us to compare with recent previous works such as **Li et. al. from ICLR’25** and **Han et. al. ICLR’23**, which was shown to perform better than another existing method Neural Diving Nair et. al.

---

> ### Author Response · Authors · 2026-04-25
> **Net-New Experiments on Backdoor Variable Prediction and Solver Configuration Prediction**
>
> The recent work from Cai et. al. 2025 (Best Paper Award at CPAIOR’25) presents a novel **multi-task learning approach** that can be trained for two tasks; backdoor variable prediction and solver configuration prediction. This is in the same spirit of ours, even though it needs to be __trained specifically on each problem domain__.
>
> With **huge credits to the authors of Cai et. al 2025**, who made their train/test splits for CA, MVC, IS problems for Backdoor Variable Prediction and Solver Configuration Prediction as well as their code pipeline available, we are able to run their approach in the exact setup that they used on these problems.
>
> We train and test their multi-tasking approach on each problem separately as suggested and we confirm that our results match their published work.
>
> To compare with this, using the same train/test splits, we take our “pretrained” Forge model and fine-tune two versions: pretrained Forge + a prediction head for Backdoors, the **same** pretrained Forge model + a prediction head for Solver Configuration.
>
> These prediction heads are similar/identical to predictions heads we used for Task 1 (a simple MLP layer).
>
> Most importantly, **we train across all problems and not perform a dedicated training per problem.**
>
> Our initial results are very encouraging. Please see the section in Appendix for details.
>
> To summarize: Forge holds potential to provide __a single pre-trained model__ that can be fine-tuned **for multiple tasks (integrality gap prediction, search guidance, backdoor variable selection, solver configuration prediction)** that __does not require per problem fine-tuning__ and improves over the best commercial solver as well as better/competitive than SOTA learning-based approaches.

---

> ### Author Response · Authors · 2026-04-25
> **Net-New Experiments Transfer Learning from Optimization to Decision Problems**
>
> To prove the generalization of Forge even further, let us go beyond the optimization literature.
>
> If Forge embeddings are indeed capturing combinatorial structure, it __should also work for other combinatorial problems__. To test this we switch entirely from Optimization problems to Decision problems, in particular **Boolean Satisfiability (SAT)**. Please see Appendix A7.
>
> We use a standard SAT-to-MIP encoding that converts SAT instances to MIP instances and take **SAT instances from G4SATBench**, a common/standard benchmark for learning-based SAT methods.
>
> Obviously, these SAT instances are __entirely unknown to Forge__. In this dataset, there are 7 different SAT problem domains each with its own SAT vs. UNSAT instances. In total, we have **14 different <problem, feasibility> pairs**.
>
> We convert these SAT instances to MIP instances using the standard encoding, and then embed them using our pretrained Forge model, which is pretrained solely on MIP instances. We repeat the same **unsupervised clustering** exercise to identify:
>
> 1) performance on differentiating across SAT problem domains, and
>
> 2) performance on differentiating across SAT vs. UNSAT instances.
>
> Our initial results, added to the revised paper as a net-new additional section at the end of the Appendix, shows that **Forge embeddings can indeed distinguish among different SAT problems and their feasibility**.
>
> Notice that, in this new experiment, our MIP specific node features are entirely __useless__ (in fact, they remain constant across all instances due how the standard SAT-to-MIP encoding works) for MIP instances converted from SAT instances.
>
> As a quick alternative, we try another setting where we replace these MIP node features with simple SAT node features (such as clause length, positive/negative ratio etc.). This version, referred to as **Forge-MIP-SAT**, still uses the same architecture and the exact same weights pretrained on MIP instances.
>
> It is extremely motivating to see that __Forge model trained solely on the MIP dataset transfer so effectively to the G4SATBench dataset__.  Please see the Appendix for the details.
>
> To the best of our knowledge, this is **the first approach for transfer learning from a pretrained optimization model to decision problems**.
>
> As a by-product, it also produces **instance-level SAT embeddings** in a fully __unsupervised manner using a pretrained model__.
>
> That means, now, with Forge, we not only get **meaningful MIP representations** but also **meaningful SAT representations** without any SAT-specific supervision or dependency on SAT solvers.
>
> Thanks to your review, this new experiment opens yet another exciting direction for us: __to train a Forge model on a mixture of MIP and SAT instances, enabling hybrid, multi-domain pretraining for combinatorial problems.__
>
> In future, It would be extremely interesting to study whether a single pretrained model can capture the structure of combinatorial optimization and satisfaction problems that works in both downstream of MIP and SAT tasks. Thank you again for your review that helped us expand our horizons!

---

### Review · Reviewer_uJoj · 2026-04-19

**Summary Of Contributions:**

The paper introduces Forge, a pre-trained foundation model for combinatorial optimisation, specifically, MIP. The authors construct the model based on a graph representation of the MIP instance. Following a 2-layer GNN, the embeddings are inputted to a vector quantised codebook, which outputs an embedding of the problem instance. Forge is trained in an unsupervised fashion, allowing training without solving the training instances, which is important, as solving MIP instances can incur large computational budget.The authors then use the embeddings for clustering of the instances, and 2 downstream tasks, where they show superior performance to other ML and non-ML baselines.

**Additional Comments:**

- Forge does not need full training data (i.e. no need for the solutions), however, it requires training instances. This raises the question: if we don’t have training instances, would it still be possible to train Forge using random, or synthetically generated instances.

**Audience:**

Yes

**Audience Explanation:**

ML venues have previously accepted multiple papers related to solving combinatorial optimisation problems using ML.

**Broader Impact Concerns:**

No ethical concerns.

**Claims And Evidence:**

Yes

**Claims Explanation:**

The authors have demonstrated the effectiveness of Forge for MIP. However, I would like to note that there are other optimisation problems, where Forge is not designed to work with. This should be made clearer.

**Requested Changes:**

- The loss function, I believe, is inspired by VQ-VAE and other foundation models. A citation should be added.

- The authors describe Forge as a general foundation model for optimisation purposes. However, they test it only on MIP instances, while other optimisation types exist, including pseudo-Boolean optimisation, constraint satisfaction optimisation, MaxSAT and more.

- In the introduction the authors say “foundational optimisation representation in an unsupervised fashion applicable to multiple optimisation scenarios”, however, later, the authors acknowledge that some fine tuning is necessary for the specific task. It would be good to clarify this sentence.

- “We use 600 instances from MIPLIB sorted by size to ensure the resulting bipartite graphs fit on GPU memory”: could you clarify how many instances were excluded? What are the sizes of the included and excluded instances? A clear limitation of such a method can be the restricted size of instances it can handle.

- The authors only compare to the Kadioglu et al 2010 hand-crafted features, and do not compare against them for the clustering task, as they cannot differentiate instances with the same numbers of variables and constraints. However, a more advanced set of features is used by Hutter et al 2014 (<https://arxiv.org/abs/1211.0906>). These features can be used even if the instances have the same number of variables, constraints etc, as they are more informative.

---

> ### Author Response · Authors · 2026-04-25
> **Response to The Loss Function Citation**
>
> Absolutely! Thank you for noticing this. We added the citation in Section where we introduced the loss.

---

> ### Author Response · Authors · 2026-04-25
> **Response to Other Combinatorial Problems**
>
> Thank you for this excellent suggestion!
>
> We have now have additional experimental sections on 2 net-new MIP tasks **Backdoor Prediction** and **Solver Configuration Prediction** with comparisons to a recently published work.
>
> Moreover, going beyond MIP tasks, we also added a net-new section on **Transfer Learning from Optimization to Satisfaction** where we use our pretrained MIP model to unsupervised SAT representations to cluster unseen SAT instances.
>
> Please see our detailed commentary to Reviewer Rsii and the new sections at the end of the Appendix.
>
> All in all, we have now showed that our pretrained Forge model can be fine-tuned for important MIP tasks; such as integrality prediction, search guidance, backdoor variable prediction, solver configuration prediction and even transfers from MIPs to SATs.

---

> ### Author Response · Authors · 2026-04-25
> **Response to Clarify Fine-Tuning**
>
> You are indeed correct that for specific downstream tasks, **fine-tuning labels are required**.
>
> An important distinction is, for both fine-tuning LP prediction and Search Guidance, we use **the same pre-trained model** for both tasks on all problems. That’s our main novelty. We clarified this at the end of Section 3.
>
> Notice also that our labeling strategy is designed to be cheap.
> * For Task - 1, labels are obtained from LP relaxations which is fast compute, and
> * For Task - 2, labels are obtained from feasible solutions found within a 2 minute time-box run.
>
> In other words, our fine-tuning uses the same pretrained model and **does not require optimal solutions or solving to optimality**.
>
> Not only that, but we use **only 50 and 100 labels** per problem for Task 1 and Task 2 respectively, and **don’t include all test problems** in fine-tuning data (e.g., MVC was never included in fine-tuning).

---

> > ### Author Response · Authors · 2026-04-25
> > **Response to GPU memory**
> >
> > Thank you for this observation. Of the 1800 MIP instances, we had to exclude ~20 instances that had more than 150K nodes due to GPU memory. Obviously this depends on the available hardware. We clarified this Section 4 - Training.

---

> ### Author Response · Authors · 2026-04-25
> **Response to more advanced MIP Features**
>
> Thank you for this remark. It is certainly possible to consider more advanced features.
>
> We are aware of the Hutter et. al. work which also includes Kadioglu et. al features, that we used here, and on top of that, quoting from their paper, it adds:
>
> `` We introduce two new sets of features. Firstly, our new MIP probing features 102 116 are based on 5-second runs of CPLEX with default settings. They are obtained via the CPLEX API and include 6 presolving features based on the output of CPLEX’s presolving phase (102–107); 5 probing cut usage features describing the different cuts CPLEX used during probing (108–112); and 4 probing result features summarizing probing runs (113–116). Secondly, our new timing features 117–121 capture the CPU time required for computing five different groups of features: variable-constraint graph, linear constraint matrix, and objective features for three subsets of variables (“continuous”, “non-continuous”, and “all”, 26–91); LP-based features (92–95); and CPLEX probing features (102–116). The cost of computing the remaining features (1–25, 96–101) is small (linear in the number of variables or constraints).
> ``
>
> In other words, these features use probing/runtime features obtained from __running a MIP Solver__.
>
> In principle, the motivation behind our work is to reduce dependency on solvers hence we do not include these types of features in our work. As also rightly noted in Hutter et. al. these features can be computationally  intensive to calculate, if not impossible for larger MIPs.
>
> We added a clarification on this in Section 4.

---

> > ### Comment · Reviewer_uJoj · 2026-05-02
> >
> > Thank you for the response. However, the feature set of Hutter et al contains 121 features from various sources ("Figure 2 summarizes 121 features for mixed integer programs (i.e., MIP instances). These include 101 features based on existing work [76, 48, 67], 15 new probing features, and 5 new timing features."). While I agree that using features based on MIP solvers is not relevant for this work, the Hutter et al feature set contains many more features which are not computed using MIP solvers, such as the graph based features and the other 91 cheap features. Therefore, a comparison to these features will strengthen the claim regarding clustering capabilities of Forge.

---

> > > ### Author Response · Authors · 2026-05-02
> > >
> > > Thank you for the suggestion - we agree this would be a valuable baseline to add. We're looking into their code base and will try to reimplement their features and report back when we have an update.

---

> > > > ### Author Response · Authors · 2026-05-07
> > > >
> > > > Thank you for your patience. We have added Appendix Section A9 that has used the ~100 features mentioned in the Hutter et al paper as a baseline comparison. All changes have been highlighted in blue.

---

> > > > > ### Author Response · Authors · 2026-05-15
> > > > >
> > > > > Thanks again to your comment, we took Hutter (2014) static mip featurizer, originally written in C++ with an older compiler version that is non-trivial to run today out-of-the-box, and ported it to a modern Python stack using the Gurobi Python interface. We open-source it as a standalone utility/helper and hope that will help others in their pipelines from MIP to Feature Vectors.
> > > > >
> > > > > I am not sharing the github link here (for anon peer review) but it's available in open-source and we can link to it later.

---

### Decision · Action_Editor_TXtr · 2026-06-02

**Recommendation:** Accept as is

**Audience:**

Yes

**Audience Explanation:**

The integration of machine learning and discrete optimization is a well-established research area with a good community of researchers. The contributions of this paper will be of interest.

**Claims And Evidence:**

Yes

**Claims Explanation:**

This paper demonstrates how their embedding for MIPs can indeed be used/fine-tuned for a set of reasonable downstream tasks. For the original submission, reviewers were concerned about the initial claim that the embedding now constitutes a foundation model, given that the set of tasks tested is not extremely comprehensive (the term "foundation model" has a very high bar). The authors agreed and toned down the claim to "a step towards a foundation model", which reviewers and I agree is now well-supported by the experimental results.

---

> ### Author Response · Authors · 2026-06-16
>
> Thank you for your time and feedback! We have uploaded the camera ready version.